# Understanding the Driving Forces That Trigger Mutations in SARS-CoV-2: Mutational Energetics and the Role of Arginine Blockers in COVID-19 Therapy

**DOI:** 10.3390/v14051029

**Published:** 2022-05-11

**Authors:** Harry Ridgway, Christos T. Chasapis, Konstantinos Kelaidonis, Irene Ligielli, Graham J. Moore, Laura Kate Gadanec, Anthony Zulli, Vasso Apostolopoulos, Thomas Mavromoustakos, John M. Matsoukas

**Affiliations:** 1AquaMem Consultants, Rodeo, New Mexico, NM 88056, USA; 2Institute for Sustainable Industries and Liveable Cities, Victoria University, Melbourne, VIC 3030, Australia; 3NMR Facility, Instrumental Analysis Laboratory, School of Natural Sciences, University of Patras, 26504 Patras, Greece; cchasapis@upatras.gr; 4Institute of Chemical Engineering Sciences, Foundation for Research and Technology, Hellas (FORTH/ICE-HT), 26504 Patras, Greece; 5NewDrug PC, Patras Science Park, 26504 Patras, Greece; k.kelaidonis@gmail.com; 6Department of Chemistry, National and Kapodistrian University of Athens, 15784 Athens, Greece; eir.ligielli@gmail.com (I.L.); tmavrom@chem.uoa.gr (T.M.); 7Pepmetics Inc., 772 Murphy Place, Victoria, BC V6Y 3H4, Canada; mooregj@shaw.ca; 8Department of Physiology and Pharmacology, Cumming School of Medicine, University of Calgary, Calgary, AB T2N 4N1, Canada; 9Institute for Health and Sport, Victoria University, Melbourne, VIC 3030, Australia; laura.gadanec@live.vu.edu.au (L.K.G.); anthony.zulli@vu.edu.au (A.Z.); vasso.apostolopoulos@vu.edu.au (V.A.); 10Immunology Program, Australian Institute for Musculoskeletal Science (AIMSS), Melbourne, VIC 3021, Australia

**Keywords:** COVID-19, SARS-CoV-2 variants, molecular dynamics, sartans, bisartans, angiotensin type1 receptor, angiotensin receptor blockers

## Abstract

SARS-CoV-2 is a global challenge due to its ability to mutate into variants that spread more rapidly than the wild-type virus. Because the molecular biology of this virus has been studied in such great detail, it represents an archetypal paradigm for research into new antiviral drug therapies. The rapid evolution of SARS-CoV-2 in the human population is driven, in part, by mutations in the receptor-binding domain (RBD) of the spike (S-) protein, some of which enable tighter binding to angiotensin-converting enzyme (ACE2). More stable RBD-ACE2 association is coupled with accelerated hydrolysis of furin and 3CLpro cleavage sites that augment infection. Non-RBD and non-interfacial mutations assist the S-protein in adopting thermodynamically favorable conformations for stronger binding. The driving forces of key mutations for Alpha, Beta, Gamma, Delta, Kappa, Lambda and Omicron variants, which stabilize the RBD-ACE2 complex, are investigated by free-energy computational approaches, as well as equilibrium and steered molecular dynamic simulations. Considered also are the structural hydropathy traits of the residues in the interface between SARS-CoV-2 RBD and ACE2 protein. Salt bridges and π-π interactions are critical forces that create stronger complexes between the RBD and ACE2. The trend of mutations is the replacement of non-polar hydrophobic interactions with polar hydrophilic interactions, which enhance binding of RBD with ACE2. However, this is not always the case, as conformational landscapes also contribute to a stronger binding. Arginine, the most polar and hydrophilic among the natural amino acids, is the most aggressive mutant amino acid for stronger binding. Arginine blockers, such as traditional sartans that bear anionic tetrazoles and carboxylates, may be ideal candidate drugs for retarding viral infection by weakening S-protein RBD binding to ACE2 and discouraging hydrolysis of cleavage sites. Based on our computational results it is suggested that a new generation of “supersartans”, called “bisartans”, bearing two anionic biphenyl-tetrazole pharmacophores, are superior to carboxylates in terms of their interactions with viral targets, suggesting their potential as drugs in the treatment of COVID-19. In Brief: This in silico study reviews our understanding of molecular driving forces that trigger mutations in the SARS-CoV-2 virus. It also reports further studies on a new class of “supersartans” referred to herein as “bisartans”, bearing two anionic biphenyltetrazole moieties that show potential in models for blocking critical amino acids of mutants, such as arginine, in the Delta variant. Bisartans may also act at other targets essential for viral infection and replication (i.e., ACE2, furin cleavage site and 3CLpro), rendering them potential new drugs for additional experimentation and translation to human clinical trials.

## 1. Introduction

Severe acute respiratory coronavirus 2 (SARS-CoV-2), the virus that causes coronavirus 2019 (COVID-19) by binding to angiotensin converting enzyme 2 (ACE2), genetically changes over time, evolving with new mutations generated across the genome, including the spike protein (S-protein) receptor binding domain (RBD) [1,2]. Most mutations have little to no impact on the viral properties; however, some are now understood to enhance or otherwise alter infectivity, transmissibility and severity of associated disease(s) [3]. During late 2020, the emergence of variants that posed an increased risk to global public health prompted the characterization of specific mutations as “variants of interest” (VOI) and “variants of concern” (VOC), in order to prioritize global monitoring and research [4]. VOCs are mainly associated with increased transmissibility and detrimental change in COVID-19 epidemiology. VOIs are genetic changes that are predicted or known to affect virus characteristics, such as transmissibility, disease severity, immune escape, diagnostic and therapeutic escape. SARS-CoV-2 variants that meet the definition of a VOC include Alpha, Beta, Gamma, Delta, Kappa, Lambda, and Omicron (see Figure 1; cf., alternative curated database of VOIs and VOCs, visit: https://en.wikipedia.org/wiki/Variants_of_SARS-CoV-2, accessed on 28 April 2022). Computational approaches implemented by us and others [5] have been applied to unravel the driving forces, on an atomic level, of the dominating mutations for SARS-CoV-2 variants that stabilize the S-protein RBD-ACE2 complex; and that can potentially retard hydrolysis of the furin cleavage site and enhance viral infection and transmissibility. It should be emphasized that while there are now >100 documented S-protein mutations associated with VOIs and VOCs (Figure 1), only a relatively small subset of these have been correlated with significant amplification of virion pathogenicity.

The aim of this in silico study was to elucidate the roles of selected mutations in SARS-CoV-2 strains, particularly regarding their ability to conformationally stabilize S-protein structure leading to enhanced RBD-ACE2 binding. The study is predicated on the knowledge that the Renin Angiotensin System (RAS), and in particular ACE2 and Angiotensin II, are implicated in the pathogenesis of COVID-19. In particular, ACE2 mediates virion attachment and entry into susceptible host cells and tissues. Viral expression and replication in COVID-19 patients is often accompanied by expression of Angiotensin II and cytokine proliferation coupled with the presentation of pneumonia and serious respiratory distress. Angiotensin II is a model hormone for ligand receptor interactions. The mechanism of this interaction, as well as those interactions of angiotensin receptor blockers (ARBs), with the AT1 receptor, are fruitful leads in understanding the mechanism of SARS-CoV-2 mutations and the mutants’ actions. In this report we describe in silico studies suggesting that certain ARBs and a new category of sartan-like drugs known as “bisartans” may offer possible treatment modalities for COVID-19. Bisartans possess a central cationic alkyl-imidazole scaffold from which extend dual anionic biphenyl-tetrazole groups. Bisartan-A (BisA) bears a butyl group at position 4 of the central imidazole ring, whereas BisD has the butyl group at position 2 (similar to Losartan). Modeling data herein are consistent with the proposition that selected mono-sartans and bisartans may interfere with viral infection and transmissibility, possibly by blocking hydrolysis of the SARS-CoV-2 S-protein.

## 2. Materials and Methods

### 2.1. Prediction of Changes in Binding Affinity of the RBD-ACE2 Complex

To calculate the structural effects of the S-protein point mutations on the RBD–ACE2 complex, the resulting model was analyzed by the MutaBind package [6]. The MutaBind method uses molecular mechanics force fields, statistical potentials, and fast side-chain optimization algorithms. For each mutation of a protein-protein complex, the MutaBind server provides the following results. (i) The free-energy (ddG) (kcal mol^−1^), predicted change in binding affinity induced by the mutation. Positive and negative signs correspond to destabilizing and stabilizing mutations predicted to decrease and increase binding affinity, respectively. (ii) Interface (yes/no). MutaBind defines a residue to be located on a protein-protein interface if the residue’s solvent accessibility in the complex is lower than in the corresponding unbound partners. To assess the impact of point mutations on protein dynamics, the complexes were analyzed by DynaMut web server [7]. DynaMut implements two distinct approaches: graph-based signatures and normal mode dynamics, which can be used to analyze and visualize protein dynamics by sampling conformations and assess the impact of mutations on protein dynamics and stability resulting from vibrational entropy changes. Computed binding energies (in kcal/mol) of AngII and sartans (i.e., candesartan, losartan, telmisartan, and BisA) on S-protein RBD-ACE2 complex were calculated based on the Mutabind and DynaMut methods (Table 1).

### 2.2. Molecular Simulations

The structure of sartans and AngII (extracted by the Protein Data Bank, PDB entry code: 4APH) were built in 3D coordinates, and their most stable (lowest energy) conformation was detected by geometrical optimization of its structure in the gas phase, as implemented in the Spartan’ 14 Molecular Modeling program suite. The structures of the molecules were initially optimized (via energy minimization) by conformational search using the Monte Carlo method with the MMFF94 molecular mechanics model included in the Spartan’14 program suite. Geometry optimization (leading to the most stable conformer with the lowest energy) was accomplished via quantum–chemical calculations utilizing the ab initio Hartree–Fock method with a 6–31 G* basis set. The structure of EXP3174 was also extracted by Protein Data Bank and its energy was minimized using UCSF Chimera 1.14. [8]. The molecular docking studies were carried out on the crystal structure of the RBD-ACE2 complex (Protein Data Bank, PDB entry code: 6LZG) [2] and the Mpro of SARS-CoV-2 (PDB entry code: 6LU7) [9]. The molecular docking simulations were performed by the open-source program AutoDock Vina included in Auto-Dock Tools 1.5.6. [7,10,11]. The best docked poses, with both lower binding energies and stronger interaction pattern, were derived from the docking results using UCSF Chimera 1.14. [8], PyMOL (Schrödinger and DeLano, 2020 PyMOL, retrieved from http://www.pymol.org/pymol, accessed on 28 April 2022) and Protein-Ligand Interaction Profiler (PLIP) [12] to visualize the molecules and the results of the docking and to construct the molecular models [13].

Molecular Dynamics simulations. MD simulations were implemented from the YASARA suite (http://www.yasara.org, accessed on 28 April 2022) according to methods previously described in detail [14]. Briefly, the main steps included optimization of the hydrogen bonding network to increase the solute stability, and pKa prediction to adjust protonation states of residues at pH of 7.4. NaCl ions were introduced to a physiological concentration of 0.9 wt%, with an excess of Na+ or Cl^−^ to neutralize the periodic cell. Following steepest descent and simulated annealing minimizations to remove clashes, the simulation was run for up to 120 ns using the AMBER-14 force field for the solute, GAFF2 [15], and AM1BCC [16] for ligands and TIP3P for water. A default AMBER cutoff of 8 Å was used for Van der Waals forces [17]. No cutoff was applied to electrostatic forces using the Mesh Ewald algorithm [18]. Using an NPT ensemble (T = 311^O^K, P = 1 atm), the equations of motion were integrated with multiple timesteps for bonded and non-bonded interactions employing algorithms described in detail previously [19].

### 2.3. Preparation of Ligands

The chemical structures of the molecules of this present study were extracted by Protein Data Bank. The charges of the molecular structures were assigned and their energy was minimized using UCSF Chimera 1.14. where the files were saved as mol2 files. In AutoDock 4.0, flexible torsions were assigned, and the acyclic dihedral angles were allowed to rotate freely. The file was then saved as pdbqt file format for further analysis.

### 2.4. Molecular Docking

The X-ray crystal structures of the SARS-CoV-2 proteins were retrieved from Protein Data Bank as pdb files, and Open Babel software was used to convert the proteins into pdbqt type [20]. Furthermore, the protein structures were refined for hetero-atoms and water molecules to demarcate active sites of proteins. The hydrogen atoms and Kollman charges were added and, subsequently, the non-polar hydrogens were merged. In each receptor, the grid was set around its active site in order for site-specific docking to be performed [21]. The Lamarckian Genetic Algorithm, which uses the AMBER force field to run the docking between the receptor and the ligand, was used for docking in combination with the grid-based energy evaluation method with default parameters (LGA: 2,500,000—energy evaluations and 150—population size). The program was run for a total number of 100 Genetic algorithm runs [22,23].

In this study, molecular docking was performed at the known active site of the Mpro (also referred to as PCLpro; PDB entry code: 6LU7), where the peptide-like inhibitor N3 is bound. The receptor grid-box was generated by AutoGrid4 with grid box dimensions of 60 Å × 60 Å × 60 Å with spacing of 0.375 Å centering around hotspot residues His41, Cys145, Ser144, Glu166, and Gln189 [9]. The grid box for S-protein RBD (PDB entry code: 6LZG), was set with spacing of 0.420 Å and dimensions of 60 Å × 80 Å × 80 Å centering around residues Lys417, Leu455, Phe486, Asn487, Gln493, and Ser494 [24,25,26,27,28]. The configuration file for the grid parameters was defined using 3D grid centres for the Mpro of SARS-CoV-2 with −10.450, 11.92, and 69.425 as X-, Y-, and Z- coordinates and for the spike protein of SARS-CoV-2, with −36.176, 31.027, and −1.875 as X-, Y- and Z- coordinates, respectively.

### 2.5. Construction of a Full-Sequence S-Protein Model for Calculation of Proteo-Stabilities

For an initial structure, a full-sequence 3D homology model (#01) was downloaded from the Swiss Model Repository at: https://swissmodel.expasy.org/repository/species/2697049 (accessed on 28 April 2022). The 3D template for this model was an electron microscope-derived structure resolved at 2.90 Å (PDB entry 6XR8). The “Global Model Quality Estimation” (GMQE) index for this model was 0.73 and the sequence similarity was 0.62. The model was subsequently annotated, refined and optimized using ICMpro (Molsoft, LLC, San Diego, CA, USA). For ddG calculations of protein stability as a function of mutant type, only the isolated A-chain of the homotrimer model was used.

### 2.6. Sartan Structures and Pharmacophores

Losartan is an ARB bearing tetrazole and hydroxyl methylene groups. EXP3174 is a losartan carboxylic acid bearing a tetrazolate and carboxylate. Olmesartan is a losartan analogue bearing a single tetrazolate and a carboxylate. Bisartans are imidazole bis-dialkylated-biphenyl-tetrazoles with the imidazole group bearing alkyl group at position 2 (losartan-like) or 4 (BisA).

## 3. Results

### 3.1. Relationship of Virion Binding to Residue Hydropathies

Enhancement of virion binding generally aligns with the trend according to the Kyte-Doolittle hydropathy scale presenting the order of amino acids from the most non-polar and hydrophobic (i.e., isoleucine; Iso) to the most polar and hydrophilic (i.e., arginine; Arg) [29] (Table 2).

### 3.2. S-Protein Free-Energy Stability Perturbations Computed for Selected SARS-CoV-2 Mutants

Eleven distinct mutation loci were of interest in this study, including A701, D80, D215, D614, E484, K417, L18, N501, P681, R246, and S982 (Figure 1 and Figure 2). The locations of these loci relative to the RBD region are depicted in the full sequence homotrimeric homology Model#05 derived from the Swiss Model Repository at https://swissmodel.expasy.org/repository/species/2697049 (accessed on 28 April 2022) (Figure 2; see Methods). The model was re-constructed and refined further in ICMpro software (Molsoft, LLC). Loci N501, Q484, S982, and K417 were of particular interest since they are situated proximate or adjacent to the interfacial region of the S-protein RBD that is intimately involved in virion attachment to the ACE2 receptor. Each of the eleven residue loci was sequentially mutated using all 20 natural amino acids. The calculation compares the single-point conformational energies of the wild-type and mutated proteins. Using the A-chain, a cross correlation matrix was generated relating mutation-driven changes in computed S-protein (A-chain) free-energy stability (ddG) values to one another. Pairwise correlations with R values > 0.5 are denoted as colored-coded double-headed-arrow dashed lines in Figure 2. Higher correlations indicate mutations at each pair locus affect proteo-stabilities to a similar magnitude. For example, sites R246 and L18 exhibited the highest correlation (R = 0.87) suggesting the possibility that mutation-driven conformational changes at one site likely influence conformation at the other site in a manner that also favors enhancement of S-protein stability.

As shown in Figure 3A, ddG values varied significantly over the eleven mutation sites, as well as by amino acid type. Note that lower ddG values correspond to increased S-protein conformational stabilities. These data indicate that, on average, Arg, Asn, Asp, Gln, Leu, and to a lesser extent, Ser and Tyr promoted S-protein stability. In contrast, Ala, Gly, His, Thr, and especially Cys substitutions generally were associated with decreased proteo-stability.

The ddG results presented in Figure 3B, in which protein stability is plotted as a function of mutational site for all 20 (color-coded) amino acids, suggest that certain mutation sites were more or less prone to stability perturbations than other sites. For example, a substantial range of amino acid type substitutions at sites D80, D215, N501 and S982 tended to increase S-protein stability, i.e., ddG values were mostly negative. However, for most other sites examined in this study, nearly any amino acid substitution typically caused a diminution of proteo-stability. At several sites, there were one or two amino acid type substitutions that promoted stability (A701, D614, E484, and L18); however, even at these sites the great majority of residue substitutions resulted in decreased proteo-stability.

In the special case of locus N501D (in which the N501Y variant is documented to be more transmissible [30,31,32], seven of the 20 amino acid substitutions resulted in an increase of S-protein conformational stability (Figure 4 upper panel). Several of the seven substitutions at the N501x site that enhanced stability included the comparatively hydrophilic amino acids Arg, Gln, Lys, and Tyr, which typically promote protein conformational stability by entering into strong electrostatic interactions and hydrogen bonding arrangements. The remainder of the N501x substitutions that enhanced stability were relatively hydrophobic residues (i.e., leucine, isoleucine, and valine). Nevertheless, these also can stabilize helical regions in particular through backbone hydrogen bonding. The tyrosine (Tyr) (N501Y) mutation is especially significant since it has the potential to enter into attractive intermolecular π-π bonding interactions at the SARS-CoV-2 RBD-ACE2 binding interface (see below, Figure 5). This observation may, in part, help explain the enhanced transmissibility/infectivity of the N501Y variant.

Interestingly, an Asn to Arg (N501R) substitution resulted in the largest enhancement of protein stability (−1.28 kcal/mol) compared not only to the wild-type (N501D) strain but also to the N501Y strain (Figure 4). Further in silico studies reported herein (see Figure 4, MD simulation results below), and previously by others [11], have shown that triple mutations in the RBD segment of the spike protein, namely N501Y/K417R/E484K and N501Y/K417N/E484K, exhibited significantly stronger binding to the ACE2 receptor compared to the wild-type RBD. At this time, it remains uncertain whether increased S-protein conformational stability consistently results in enhanced viral transmissibility and/or infectivity. However, such a relationship does appear to exist at certain mutational sites, such as N501x, which when mutated to Y is known to be a more transmissible variant relative to the original SARS-CoV-2 wild-type strain.

### 3.3. Binding Free-Energy Calculations between SARS-CoV-2 RBD and Human ACE2 Receptor

Previous studies have shown that RBD double mutations of SARS-CoV-2 strains increase transmissibility through heightened interactions between RBD and ACE2 receptors [33]. Single and double mutations in the RBD segment (residues 318–536) of the S-protein (residues 1–1273), namely, N501Y (alpha), E484K/N501Y (beta, gamma), L452R/T478K (delta), L452R (epsilon), L452R/E484Q (kappa), and L452Q (lambda), enhance binding of the mutated RBD to the ACE2 receptor via attractive electrostatic and π-π resonance interactions, as predicted and demonstrated in in-silico studies [5,33]. Tyr, even less polar and hydrophilic compared with Asn in the N501Y mutation, offers more possibilities for stronger interactions with ACE2 residues due to the aromatic ring and the tyrosinate anion which creates strong π-π interactions, salt bridges and hydrogen bonds [34,35,36].

Changes in binding ddG between the SARS-CoV-2 RBD and the human ACE2 receptor complex (PDB: 6LZG) were calculated for 20 amino acid substitutions (single-letter labels in the graphic, (Figure 5) at three specific mutation loci, including N501x, K417x, and E484x. The method, invoked from within ICMpro software (Molsoft, LLC), computed the binding free-energy change of a protein complex (dG_bind_) as a difference between the free-energy of the mutant and wild-type forms:$$\Delta \Delta G_{bind}=\Delta G_{bind}^{mut}-\Delta G_{bind}^{wt}\;,\;where:\;\Delta G_{bind}=\left(E_{intra}^{comp}-E_{intra}^{parts}\right)+\left(\Delta G_{solv}^{comp}-\Delta G_{solv}^{parts}\right)\;$$
where *E^comp^* represents the energy of the complex, and *E^parts^* represents the sum of the energies, internal or solvation, for each part. The energy is calculated for a fixed backbone and all side chains except those in the vicinity of the mutatable residue. Prior to calculation, Monte Carlo simulations were performed to relieve atomic clashes resulting from mutations to larger residues.

More negative values in the graph (Figure 5) correspond to more robust protein-protein binding. For example, the N501H (His) mutation resulted in the greatest RBD-ACE2 binding stability at this locus (−0.95 kcal/mol). Phe (N501F, −0.09 kcal/mol) and Gln (E484Q, −0.29 kcal/mol) mutations also promoted RBD-ACE2 binding, albeit to a lesser extent. The largest calculated gain in binding stability (approximately −1.50 kcal/mol) was associated with Arg substitutions at the K417x and E484x loci. As expected, the majority of mutations introduced at these three loci resulted in significant conformational destabilization of the RBD-ACE2 association. Comparison of dG_bind_ values of the RBD mutants N501F, N501Y, and N501W revealed the latter resulted in the largest destabilization of the RBD-ACE2 intermolecular interaction (+1.58 kcal/mol) among the 20 single-point mutations examined (see Figure 5). However, this apparent weakening of binding inferred from the dG_bind_ calculations for the Trp substitution (N501W) stands in contrast to binding energy results extracted from the MD trajectories (see Figure 6 and Figure 7). The MD simulations predicted the N501W mutation should theoretically enable an increase in binding strength of the RBD-ACE2 complex. As alluded to below (Section 3.4), this discrepancy is likely due to significant differences in the computational approaches (dG_bind_ vs. MD). Notwithstanding such differences, it seems plausible to expect more accurate binding energy predictions from MD simulations that better account for critical solvation effects through the use of explicit solvent models, as well as local and global conformational dynamics, possibly resulting in optimization of the initial x-ray structures. This line of reasoning is consistent with the observation that, in our MD simulation models, Trp, like Tyr, can clearly enter into attractive π-π resonance interactions with the Y41 aromatic ring of the ACE2 receptor, thereby increasing RBD-ACE2 binding (see Figure 7, below).

Based on the dG_bind_ calculation results presented in Figure 5, and assuming a positive correlation between RBD-ACE2 affinity and virion transmissivity/infection, we may extrapolate higher transmission rates (relative to the wild-type) for variants possessing one or more of the following mutations: N501F, N501H, N501M, N501Y, E484Q, E484R or K417R, all of which exhibited gains in ACE2 binding affinity, i.e., negative dG_bind_ values. As indicated in Figure 1, some of the RBD mutants, including both N501Y (Alpha, Omicron) and E484Q (Delta, Kappa) are, in fact, associated with more rapid spread of SARS-CoV-2. However, most other mutations examined, including N501H that exhibited the largest gain of binding energy for this locus (dG_bind_ = −0.95 kcal/mol, Figure 5), have not, to our knowledge, been routinely or commonly observed in actual variant sequences. Nevertheless, based solely on the free-energy perturbation calculations in Figure 5, a prediction could be made that such mutants may eventually be discovered in humans, and could even become dominant due to their ability to undergo stronger binding to the ACE2 receptor.

### 3.4. MD Simulations: Binding Energies between SARS-CoV-2 RBD and Human ACE2 Receptor

Molecular dynamics simulations have been employed extensively over the past several years to probe SARS-CoV-2 binding motifs [37]. Herein we too make use of fully atomistic MD simulations to calculate and compare RBD-ACE2 binding energies for the 6LZG complex. Binding energies were computed from 12–18 ns MD trajectories for the wild-type RBD-ACE complex, as well as for ten RBD mutant complexes, including N501H, N501M, N501Y, N501F, N501W, E484K, K471N, K417R, and two triple (3x) mutants: N501Y + K417R + E484K, and N501Y + K417N + E484K) (see Figure 6). All MD simulations were performed in a physiological saline environment (0.9 wt% NaCl) with explicit solvation (TIP3P model) at a constant temperature of 311^O^K using AMBER-14 force field parameters and an NPT ensemble (see Methods). Binding energies were adjusted for solvent-induced effects using a boundary-elements method to approximate the entropic cost of exposing 1.0Å^2^ to the solvent in kJ/mol [Yasara.org].

All of the RBD mutants examined in this study resulted in enhanced binding to the ACE2 receptor compared to the wild-type RBD, a result that is consonant with the observation that mutations at these loci are associated with heightened viral transmission. Furthermore, compared to the wild-type RBD, both “3x” mutants exhibited significantly stronger binding in MD simulations (by about 150 KJ/mol) to the ACE2 receptor (PDB: 6LZG). Thus, it may be reasonably hypothesized that these mutations acted cooperatively or synergistically to enhance RBD-ACE2 binding.

With the exception of N501W, most of the mutations that elicited enhanced RBD-ACE2 binding as evaluated by the dG_bind_ calculations (Figure 5) also exhibited qualitatively stronger binding in the MD simulations. The discrepancy between dG_bind_ and MD-derived binding energies for N501W may not, however, be surprising given that the former method (dG_bind_) represents a highly theoretical approach based solely on a static x-ray crystallographic structure of the protein. This is quite different from equilibrium MD simulations in which the system is allowed to undergo considerable conformational relaxation (energy optimization) in a fully solvated and pH-controlled environment prior to computing the protein-protein interaction energies.

In the case of N501W, the tryptophan orientation was stabilized by two intrachain hydrogen bonds, one involving the hydrogen atom of the aromatic nitrogen of W501 bonded to the carbonyl oxygen of G496, and the other involving the hydrogen atom of the amide nitrogen of W501 bonded to the carbonyl oxygen of Q498 (Figure 7). Additionally, attractive intermolecular tilted-ring π-π interactions were observed following 9ns of MD simulation between Y41 of the ACE2 receptor and phenylalanine (N501F), tyrosine (N501Y), and tryptophan (N501W) substitutions. Depending on the precise orientation and separation distance of the interacting aromatic rings (e.g., “T”, “Sandwich” and “Parallel forms”, see Figure 7F), such π-π intermolecular interactions exhibit association energies in the range of a 1–10 kJ/mol, and are known to contribute significantly to protein quaternary structure stabilization. Thus, it is reasonable that the π-π interactions observed for these RBD mutants (i.e., N501 F,Y,W substitutions) could play a key role in enhancing the binding (and hence transmissibility) of the SARS-CoV-2 virus to host cell ACE2 receptors.

The enhanced proteo-stability imparted by N501Y was consonant with a decrease of −0.36 kcal/mol of the RBD-ACE2 dG_bind_ compared to that of the N501 wild-type complex (Figure 5). In contrast, the isolated single-point mutations E484K and K417N were by themselves proteo-destabilizing, with ddG values of +0.5049 and +0.9442 kcal/mol, respectively (Figure 3); and K417N also exhibited a positive dG_bind_ value of +1.63 kcal/mol (Figure 5), suggesting it weakened RBD-ACE2 binding.

However, in MD simulations, both mutants (i.e., E484K and K417N) exhibited improved binding to ACE2. As noted above, MD simulations also revealed that when the N501Y, K417N and E484K mutations coexisted in the RBD (see 3xMut1, Figure 6), their combined effect was to significantly increase the RBD-ACE2 binding potential above that of the wild-type or any of the three single-point mutations (Figure 6).

The discrepancy between dG_bind_-derived and MD-derived binding values for the aforementioned mutations is most likely a result of the different computational approaches (i.e., dG_bind_ vs. MD) for evaluating protein-protein binding free energies. We maintain the later (MD) approach can be expected to be more consistent and accurate, since it accounts for explicit solvation (entropic) effects, as well as dynamical conformation changes averaged over the MD trajectories.

### 3.5. Steered Molecular Dynamics: RBD-ACE2 Dissociation Behavior

Steered molecular dynamics (SMD) can simulate a vectored, i.e., non-random directional force (or acceleration) needed to dissociate two bound proteins, such as the RBD-ACE2 complex (PDB 6LZG) used in this investigation. The technique has been applied in recent investigations of force-induced insulin dimer dissociation [38] and SMD simulations of irisin dimers [39].

We carried out SMD simulations on the wild-type 6LZG complex and three mutant forms, including N501Y, N501W and 3xMut1 harboring N501Y, K417N and E484K. The results of the SMD simulations are presented in Figure 8.

The SMD simulations were commenced following a series of initial Monte Carlo annealing and MD steps to equilibrate/relax the system using similar conditions to normal MD (see Methods; Section 2.2): explicit (TIP3P) solvation in physiological saline (0.9 wt% NaCl, pH 7.0), constant temperature of 311^O^K, AMBER14 parameters, multiple timesteps of 1.25 femtoseconds for intramolecular and 2 × 1.25 fs for intermolecular forces, periodic boundaries with 8 Å cutoff and long-range particle-mesh Ewald electrostatics. The major differences, however, were use of an elongated periodic cell (x-axis: 275 Å) and a bias acceleration applied to the RBD atoms to gradually pull the protein away from the tethered ACE2 receptor. The pulling force was applied along the normalized vector (center-of-mass line) pointing from the epicenter of the ACE2 receptor to the epicenter of the bound RBD of 6LZG (Figure 8A). The SMD simulations were implemented using a modified Python algorithm (Yasara.org). The vectored acceleration was gradually increased starting at 10 pm/ps^2^ until the proteins began to separate. When the separation motion was stalled for a predetermined period (due to attractive protein-protein interactions that could not be overcome at the current acceleration), the acceleration was advanced by 5 pm/ps^2^ increments. This method allowed an estimation of the “minimum pulling force” (MPF) required for complete separation of the bound proteins (Figure 8). The MPF values were corrected by subtracting a “background” force needed to continually drive the RBD through the solvent phase in the absence of ACE2. The background force was determined independently for the wild-type and each of the three mutant strains.

Several frame-captures from the SMD simulation of the wild-type RBD-ACE2 complex (6LZG) are shown in Figure 8A. Frame-captures of the mutant complexes are not shown since they appeared very similar to the wild-type complex at the journal print resolution. Surprisingly, the SMD simulations revealed that the region of the complexes proximal to RBD residue N501 and ACE2 Tyr41were the first to undergo complete dissociation as the pulling force was progressively increased along the center-of-mass axis. This initial separation occurred at about 60–70 ps into the simulations. The last region of the interfacial space to undergo separation was located distal to the RBD N501 locus in the region of the Ser19 residue at the N-terminus of the ACE2 receptor. This area involved numerous intermolecular interactions between the RBD and ACE2, encompassing RBD residues 474–490 and *alpha*-helical ACE2 residues 19–22 and 78–83 (see Figure 8A and Figure 9). This region began to completely dissociate in the wild-type complex after approximately 160–175 ps of elapsed time.

As noted in Figure 8B and Figure 9B, the MPF needed to complete separation of the wild-type RBD from ACE2 was about 3265 pN, and this force was attained when the vector separation distance was approximately 7.4 Å. Compared to the wild-type complex, all of the RBD mutant forms exhibited higher MPF values: 3642 pN for N501Y, 4001 pN for the 3xMut1 triple mutant, and 4376 pN for mutant N501W. These data qualitatively agree with the MD simulation findings discussed above (see Section 3.4) correlating stronger RBD-ACE2 associations with the occurrence of key single or multiple S-protein mutations found in common SARS-CoV-2 variants.

It should be noted that the MPF (pN) values reported herein are substantially larger than those reported in some other computational and experimental studies [40,41]. Protein dimer dissociation values typically cited range from about 100 to 1000 pN. The MPF values referred to in the current study were required to fully disengage the RBD-ACE2 heterodimer over the duration of the SMD runs. However, the “initial dissociation force” (IDF) at which time actual protein-protein dissociation was observed to commence (see Figure 8A) was considerably lower than the MPF values, generally about 200 to 500 pN, which is well within the range cited in many other publications. Moreover, the duration of the SMD simulations were a few hundred picoseconds, and it is unknown if longer duration SMD runs (e.g., ns or ms) could have eventually resulted in complete dimer dissociation at the much lower IDF values.

These results infer that the protein-protein interfacial domain located in the immediate region of the Ser19 N-terminus residue of the ACE2 receptor (i.e., furthermost from residue N501 of the RBD) is mainly responsible for RBD-ACE2 binding affinity. Closer inspection of this “strong binding domain” (SBD) in our models revealed the ACE2 side of the SBD is dominated by the two parallel *alpha*-helical regions that contribute stabilization of the SBD primarily through hydrogen bonding arrangements (Figure 9D). These *alpha*-helices, one of which harbors the Ser19 N-terminus of ACE2, overlay the central zinc-pocket cleft of the ACE2 receptor. There was also a strong delocalization π-π interaction (via a parallel-displaced configuration; see Figure 8E) between Phe486 of the RBD and Tyr83 of ACE2. Residue Asn48 of the RBD was involved in two hydrogen bonds to the -OH group of the ACE2 Tyr83 residue, as well as the terminal carbonyl group of Gln 24 of the ACE2 receptor. A strong hydrogen bond between Ser35 of ACE2 and Ala475 of the RBD also contributed to stabilization of this interfacial domain. It is this particular hydrogen bond that was among the last intermolecular interactions to undergo dissociation in DMD simulations.

### 3.6. Modeling the Role of Arginine and the Charge Relay System

In silico rational drug design is of vital importance in the process of drug discovery. It is essential to screen and test drugs for their possibility to act as effective ligands on specific targets. Molecular docking studies are of great importance to discover potential drugs against COVID-19, which involves the cardiovascular system [42,43,44] as infection is initiated by the entry of the S-protein via ACE2, a peptide of the renin-angiotensin system (RAS). Crystal structure studies have shown interactions of EXP3174 and ARBs with AT_1_R residues (Arg167, Lys199, Tyr35, and Trp84). Docking studies with SARS-CoV-2/ACE2 complex and S-protein hydrolysis entries have also shown critical interactions, rendering them potential drugs for treating COVID-19. In particular, AT_1_R Tyr35 hydroxyl binds to the imidazole nitrogen and AT_1_R Arg 167 binds to the anionic tetrazolate and carboxylate groups of Olmesartan (a commercially available ARB) through a inter-charge relay system (CRS) mechanism [45,46] (Figure 10A,B). Intra-CRS mechanism interactions were postulated in AngII and serine proteases (Figure 10C,D), and CRS intermolecular interactions with AT_1_R Arg 167 and Tyr 35 are depicted in Figure 10E,F. Docking studies of the mono-sartan EXP3174 with SARS-CoV-2 entry points (furin basic cleavage site and 3CLpro cleavage site) reveal strong interaction of the ARB tetrazole group with S-protein arginine residues inferring they are potential drugs for treating COVID-19 (Figure 11). Figure 12 shows EXP3174 incorporated in cavities of Mpro of SARS-CoV-2 and the S-protein RBD. Bisartans, discovered and synthesized in our laboratories, bear two anionic biphenyl-tetrazole groups [47,48] that could either block RBD arginine and lysine mutants (Delta variant) from binding to ACE2 acidic residues or prevent hydrolysis of the furin-rich Arg cleavage site 681–686 and the S-protein by 3CLpro, which both trigger infection [49,50,51,52]. Arginine, bearing a positive charge distributed over the three nitrogen atoms of the guanidino group through resonance mechanisms, binds strongly to anionic groups such as tetrazolates or carboxylates, which represent important pharmacophore groups in sartans and related drugs. Such electrostatic interactions could potentially block RBD binding with the ACE2 receptor; or function as a catalyst inhibitor of the furin cleavage site (Figure 13 and Figure 14).

### 3.7. Normal-Mode Molecular Dynamics

Residue E484 (Glu484) is located at the RBD-ACE2 binding interface (Figure 9A). Based on the X-ray RBD-ACE2 complex (PDB: 6LZG), negatively charged Glu484 of RBD makes strong electrostatic contacts with positive K31 (Lys31) of ACE2; thus, its substitution to arginine or glutamine destabilizes the protein complex leading to positive dG_bind_ values (+0.21 kcal/mol). The point mutation L452R and its combination with T478K have a positive impact on the RBD-ACE2 binding affinity (negative dG_bind_ numbers: −0.3 kcal/mol). MD based on normal mode approaches revealed that L452R mutation destabilizes and increases the flexibility of the RBD loop that interacts with ACE2 (Figure 9B). This is probably due to the loss of critical hydrophilic contacts of leucine after its replacement by Arg (Figure 9C,D). The increased dynamics of the RBD loop could drive conformational changes that favor electrostatic interactions between Arg452 and the positively charged alpha-helical region of ACE2 (Glu 25 and 37, Asp38) and thus the binding affinity of the protein complex. AngII, telmisartan, losartan, candesartan in the interface between ACE2 and spike protein are depicted in Figure 9 lower panel A–D.

### 3.8. Molecular Interactions of EXP3174 with Mpro and Spike Protein

Molecular docking using AutoDock VINA of EXP3174 provided pdbqt files that were converted to pdb using Open babel. Then, the pdb files were analyzed by PyMOL and PLIP. The in silico experiments between the proteins and EXP3174 were repeated three times for each ligand and the average values for binding affinity were recorded. The deviation of each compound was around ±0.5 on the binding energy.

Mpro of SARS-COV-2 (PDB entry code: 6LU7) displayed an excellent docking score of −10.22 kcal/mol with EXP3174. EXP3174 formed seven hydrogen bonds with Leu141, Ser144, His163, His172, two with Thr190 and one with Gln192. Three hydrophobic interactions were also observed with Met165, Glu166, and Pro168. Apart from these, one π-stacking with His163 and one halogen bond with Gln192 was formed (Table 3).

S-protein of SARS-COV-2 (PDB entry code: 6LZG), displayed an even higher binding energy (−10.91 kcal/mol) with EXP3174. In this case, EXP3174 formed six hydrogen bonds with Arg393, two with Arg403 and one with Ser494, Gly496 and Tyr505. However, five hydrophobic interactions were formed too, with Asn33, two with Glu37, one with Lys417 and one with Tyr505. Lastly, also observed were one π-cation interaction with Arg393 and one salt bridge with Lys353 (Table 4).

3D representations showing the interactions of EXP3174 with Mpro and the S-protein RBD of SARS-CoV-2 using PyMOL are depicted in Figure 7.

Figure 14 shows a closed view of the docking pose of bisartan A in the furin cleavage site with wt ACE2, P681R ACE2, and P681H ACE2, and also a closed view of the docking pose of EXP-3174 in the furin cleavage site with wt ACE2, P681R ACE2, and P681H ACE2. Figure 9 shows Charge Relay System Protease mechanisms. The SARS-CoV-2 S-protein can be cleaved by furin at positions 685–686 and 3CLpro at glutamine positions by charge relay system mechanisms. The catalytic center of furin is the triad Asp—His—Ser, and for 3CL protease is the dyad Cys145—His41. [53]. To assist us in portraying the type of bonds correctly, we used the Protein-Ligand Interaction Profiler [12]. Figure 12 shows the topomer of EXP-3174 and also shows the topomer of bisartan A and its docking with the basic furin cavity site 681–686. Figure 13 shows RBD 452 contacts (R,L) with ACE2 residues (Upper panel) and the binding of angiotensin II and ARBS (Telmisartan, Candesartan, Losartan) in the ACE2 zinc domain groove complexed with S-protein’s RBD (Lower panel). Figure 14 shows closed view of docking pose of bisartan A in arginine rich furin cleavage site. Figure 15 shows Charge Relay System protease mechanism (CRS).

## 4. Discussion

This in silico study focuses on the atomic and molecular properties of SARS-CoV-2 mutants to investigate the energetic driving forces that trigger mutations and increase transmissibility of infection. These studies have shown that arginine blockers such as bisartans, compounds that bear two anionic tetrazole groups, can block efficiently interactions between the RBD and ACE2 receptor as well as possibly interfere with the hydrolysis of the furin and 3CLpro cleavage sites that trigger infections. Hydrogen bonds, salt bridges and π-π interactions contribute to enhanced binding of selected RBD mutants with the ACE2 receptor. The application of non-equilibrium SMD simulations in this study qualitatively agreed with conventional MD results in terms of predicting which RBD mutants undergo strong binding to ACE2. The SMD simulations additionally revealed a region surrounding the Ser19 N-terminus of the ACE2 receptor that exhibited enhanced RBD binding. Referred to as the “strong binding domain” (SBD), this is the region of the RBD-ACE2 complex that was last to fully dissociate in the SMD simulations. Judicious mutagenic engineering of the SBD could possibly play a role in the improvement of “ACE2 traps” as therapeutics for COVID-19.

### 4.1. Dominating Mutations

ACE2 of the RAS is the entry of SARS-CoV-2 in healthy host cells and causes infection. The crystal structure of the RBD S protein/ACE2 complex revealed critical interactions which link the two chains [1]. This binding is strengthened by mutations that stabilize the S RBD structure of the complex formed from the S protein RBD with the ACE2 enzyme. The major coded critical mutations detected initially to occur in wild-type spike protein are N501Y, K417N, E484K, P681H, D614G, with the first, N501Y, dominant in the UK or Alpha variant, to prevail. A recent variant called Delta, with mutation L452R dominating, is more infectious and prevails over Alpha, Beta, and Gamma. The RBD 501Y residue is linked with ACE2 Y41, which furthermore strengthens the binding RBD/ACE2 compared to wild type RBD N501/ACE2 Y41 and increases infectivity and transmissibility. The Delta L452R mutation strengthens the binding between RBD and ACE2 compared to the Alpha N501Y mutation and consequently is more infectious [30,31,32]. Disruption of binding is a key to treatment therapeutics. Dominating mutations Alpha (N501Y, E484K, K417N), Beta (K417N, E484K), Gamma (K417N, E484K), Delta (T478K, L452R, P681R, N501R, E484R), and Lambda (E484Q) reveal the tendency of non-polar and hydrophobic amino acids in the wild spike protein to be replaced by more polar and hydrophilic residues (Table 2) that bind more strongly with ACE residues. The Delta variant with dominating mutations T478K, L452R, P681R, and E484R is seen to be emerging as an aggressive and more infective pandemic due to stronger binding with ACE2 residues. A common feature of the delta four mutant amino acids (i.e., 478K, 452R, 484R, and 681R) is the positive charge in the side chains of lysine and arginine. The positively charged groups, amino (K) and guanidino (R), could interact with negatively charged groups of the ACE2 enzyme-like E and D carboxylates to form strong bonds through salt bridges. Furthermore, P681R at the furin cleavage site increases its basicity, which enhances the rate of cleavage and thus infectivity. Overall, arginine, which is a highly hydrophilic, polar, and strongly positive amino acid (Table 2) seems to be the critical amin acid for the high affinity of SARS-Cov-2 with ACE2. The strong binding of arginine RBD mutants with ACE2 negatively charged amino acids such as E and D, and the enhancement of the basicity of the cleavage site 681–686, suggest the development of arginine RBD/ACE 2 blockers. If the RBD E484R mutation gains dominance it will be life-threatening, especially if it is coupled with a possible N501R mutation where arginines could further strengthen the RBD/ACE binding.

### 4.2. Mutations and π-π Interactions

The intermolecular interactions of the RBD-ACE2 complex reveal the underlined forces that drive mutations as follows. First, based on the MD modeling study, the singular π-π interaction of the two tyrosine residues (spike N501Y and ACE2Y41 in N501Y) is what really drives the binding (Figure 4 upper panel). There is, indeed, a weak H-bond between N501Y-RBD and Y41ACE2, but it is quite distant (>2.5 Angstroms) compared to the other H-bonds [1]. The single π-π interaction would overwhelm the weak H-bond in terms of the interaction potential. Second, due largely to the π-π interaction, the N501Y interaction with Y41 of ACE2 is the closest atom interaction (about 1.6 Angstroms) of all of the interactions detected by the modeling. This proximal association is most likely due to the π-π interaction and not to the weak H-bond, although the latter could indeed make a partial contribution by withdrawing electron density from the ring(s). The π-π interaction could assume slightly different conformations following repeated energy optimizations, e.g., T-form, Sandwich, or Parallel (Figure 4 lower panel). Application of state-of-the-art ab initio electronic structure theory methods elucidated the nature of π-π interactions in the benzene dimer. This study predicts the T-shaped and parallel-displaced configurations of benzene dimer to be nearly isoenergetic [54,55].

### 4.3. Targets for SARS-CoV-2 Treatment

A target for SARS-CoV-2 treatment could be the binding domain between SARS-CoV-2 and ACE2. In the crystal structure study by Lan, J. et al. [1], multiple Tyr residues (Tyr/449, 489, 505) in the RBD of SARS-CoV-2 interact with ACE2 to residues D38, Y83, R393, strengthening the binding between virus and ACE2. Other hydrogen bonds and salt bridges in the interface between SARS-CoV-2 and ACE2 were observed for interacting residues namely Asn/Gln, Lys/Asp, Gln/Glu, Tyr/Glu (and Asp), Thr/Tyr, Asn/Tyr, Asn/Tyr, Gly/Gln, Tyr/Gln, Tyr/Gln, Tyr/Tyr, Asn/Tyr (and Gln, Glu, Asn), Gly/Lys, Tyr/Arg, Lys/Asp, Lys/Glu. These interactions increase the avidity of the SARS-CoV-2/ACE2 binding by stabilizing the charging network [1,50,51,52,56]. In silico studies and clinical findings have indicated ARBs as a potential treatment of SARS-CoV-2. Furthermore, ARBs bind to ACE2 and SARS-CoV-2 critical residues, thus inhibiting the entry of RBD S protein to the enzyme.

### 4.4. P681R Delta vs. P681 Version

P681R S-protein mutation enhances the replication efficiency of the SARS-CoV-2 Delta variant [57]. Given the significance of the furin cleavage site in viral entry, the P681R mutation might enhance the infectivity of the delta variant by increasing S1/S2 cleavage at the furin site. For the validation of the significance of P681R for enhanced infectivity, a wild type P681 Delta version was compared with the mutated R681 variant. It was found that the delta version carrying wild type P681 has significantly lower replication efficiency than the original Delta variant with P681R mutation. This is in line with findings that the rate of furin cleavability is increased and, therefore, infectivity, as the basicity of the furin site is increasing from wild type P681 to His681 and then to Arg681. Interestingly, in both delta and omicron variants, the furin mutations in the cleavage site 679–686 lead to more basic mutants, which are P681R for Delta and P681H and N769K in omicron.

Docking studies revealed that bisA binds to the furin cleavage site of wt, P681R, and P681H of ACE2 with 6.3 kcal/mol, 6.5 kcal/mol, and 6.4 kcal/mol binding energies, respectively (Figure 10). In the wt and mutated ACE2 furin site, the tetrazole group of bisA makes strong contacts with Arg683. In the P681H, one of the two tetrazoles makes additional contacts with the imidazole ring of His. EXP-3174 binds to the furin cleavage site of wt, P681R, and P681H of ACE2 with 5.5 kcal/mol, 6.5 kcal/mol, and 5.6 kcal/mol binding energies, respectively (Figure 6). In the wild-type and P681H, tetrazole of EXP-3174 makes contacts with Arg682 and in P681R contacts between tetrazole group and Arg681, and the carboxyl group of EXP-3174 and side chain of Arg681 were identified.

### 4.5. CRS Mechanisms in AngII and SARS-CoV-2 Proteases

The octapeptide AnII (DRVYIHPF) acts on the AT_1_R and is a model for ligand receptor interaction. The interaction of AngII with its receptors involves a CRS mechanism analogous to serine proteases [58,59]. Cardiovascular disease is related to COVID-19 in terms of mechanisms that trigger the disease [42]. For example, the storm of cytokines released in COVID-19 patients with pneumonia is related to the over-expression of toxic AngII in the RAS [60,61]. The initial step of SARS-CoV-2 infection is the binding of the RBD in its S protein to the ACE2 receptor resulting in the hydrolysis of S protein either with furin enzyme at the basic cleavage site 681–686 or with 3CL protease at glutamine positions triggering infection [62,63]. Tyrosinate in AngII, serinate in furin, and cysteinate in 3CLpro are anions created by the CRS mechanism and trigger activity via their nucleophile anions. CRS anions (tyrosinates, serinates, cysteinates) can be blocked by anionic groups such as tetrazolates and carboxylates in ARBs, and nitrile, the warhead of antiviral drug PF-07321322 [59,62,63,64,65], which destabilizes the CRS of AngII and of the proteases, inactivating their action at the receptor level.

### 4.6. ARBs Could Bind to the Basic Arginine Rich Cavity Loop of SARS-CoV-2

The multi-basic cleavage site, P681R682R683A684R685S686 of the S1 subunit is essential for fusion of the virus with ACE2, and is an inhibition target [51,52]. This basic cavity loop is reported to be the critical sequence in the S1/S2 cleavage site of the virus, which is between R685/S686 amino acids (Figure 6). Subunit S1 contains the RBD and subunit S2 contains the fusion peptide. The S protein of SARS-CoV-2 harbors the special S1/S2 furin-recognizable site, the entry to ACE2, which is the proteolytically sensitive and treatment target. It is of note that the globally prevailed P681H (Proline681Histidine) mutation is within the fusion region, increasing the number of basic residues to four, resulting in slightly increased cleavability [66]. Proline is a rigid amino acid extensively investigated in AngII conformational studies giving rise to cis-trans isomers [67,68]. Proline changes the direction of a peptide sequence, with the restricted ability to participate in charge networks, as it lacks polar groups. On the contrary, histidine is a polyfunctional amphoteric amino acid, which can behave as acid and base and can form π-π interactions with aromatic residues. This allows tighter binding of histidine with neighboring charged groups or with aromatic residues, and participation in the CRS, as in AngII and serine proteases. The polyfunctional role of histidine is clearly depicted in these systems where it plays a central role to mediate between two interface amino acids, as proton acceptors leading to tyrosinate in AngII and serinate in serine proteases, through charge relay system mechanisms [58,59,69].

A mutation at P681R in the cleavage site 681–686 (PRRARS) has been observed in the Delta variant increasing the number of arginines (RRRARS) and further facilitating cleavage. The P681R mutation, a hallmark of the Delta variant, enhances viral fusogenicity and pathogenicity [57]. It has been reported that the lower the number of arginines in the cleavage site, the less the severity of infection. Arginines catalyze the cleavage reaction by contributing to the formation of serinate, as histidine does, in cleavage through a CRS mechanism in furin involving the triad Asp-His-Ser [58]. Arg blockers, such as Arbs or bisartans, could be promising treatments for COVID-19.

### 4.7. ARBs as Potential RBD/ACE2 Blockers

Repurposable drugs interfere in the RBD-ACE2 interface disrupting binding and consequently decreasing infectivity and transmissibility could be ARBs, which are well-known Arg blockers [70]. One representative example is the sartan, olmesartan (Figure 5) that interacts with the K199 and R167 amino acids of the AT_1_R through its tetrazolate and carboxylate. In this complex, the binding is strengthened with a hydrogen bond between Y35 of AT_1_R and Olmesartan imidazole and also with π-π interaction of the receptor rings Y35, W84 with imidazole [45,71,72]. In particular, the cationic guanidino side chain of R167 of the receptor forms a salt bridge with the carboxylate and tetrazolate of ARB olmesartan [45,46,71]. These interactions reveal a unique network of charged interactions between ARBs and AT_1_R. In silico docking studies suggest that ARBs are anchored in a deep groove in ACE2 (Figure 8) with comparable scores to those of AngII (Table 1) and could act as allosteric inhibitors of the RBD-SARS-Cov-2 complex. Among ARBs, bisartanA and telmisartan have the highest binding energy, followed by Losartan and Candesartan (Table 1). Clinical studies have shown that ARBs reduce morbidity and mortality in hypertensive patients infected by SARS-CoV-2 [73,74,75,76].

### 4.8. Bisartans and the Role of Tetrazole

We have developed a new generation of ARBs, called bisartans, the structures of which include two biphenyl tetrazoles [47,48,77]. Bisartans bear a butyl substituent at position 2 (losartan-like) or at position 4 (bisartan A) (Figure 6). These structures resulted from our structure-activity studies on AnII [47,59,69,78]. Bisartans are potent antagonists of AT_1_R and contain powerful tetrazoles, chelators of the zink proteases such as ACE2 and Neprilysin (NEP) [79,80,81]. The Delta variant with dominating mutations T478K, L452R, P681R and, in particular, P681R is an aggressive and more infective pandemic virus strain due to stronger binding of the S protein with ACE2. A common feature of the Delta mutant K and R amino acids (i.e., 478K, 452R, 484R, and 681R) is the positive charge in the side chains of lysine and arginine. The positively charged groups, amino (K) and guanidino (R), interact with negatively charged groups of the ACE2 enzyme-like E and D carboxylates to form strong bonds through salt bridges (Figure 8 upper panel). Bisartan A binds with the acidic amino acids in the ACE2 open channel close to the zinc-binding motif (Figure 11). The observed interactions are: (i) π stacking between Tyr202 and the aromatic group of bisartan; (ii) H-bond between OH group of Tyr202 and N atom of bisartan; and (iii) salt bridges between the second tetrazolate with Arg514 of ACE2. Bisartan A also binds to ACE2 where positions Leu452 and Thr478 in S protein RBD are replaced by Arg and lysine, respectively, in the Delta variant close to the nearest negatively charged residues (E22, D38, and E35) in ACE2, with which they could form salt bridges (Figure 11). Furthermore, the Delta S mutation P681R in the polybasic cleavage site 681–686, containing three arginines, enhances the cleavage of the full spike protein between 685–686 positions by furin enzyme to S1 and S2 subunits leading to increased infection via cell surface entry. The S1 fragment binds to the ACE2 receptor and the S2 fragment interacts with the membrane [82]. The more polybasic the S1/S2 cleavage site, the more infective the virus [52]. The mutations in RBD and the cleavage site with replacement of residues by arginines are treatment targets and the key to developing arginine blockers to decrease spike affinity to ACE2 and attenuate infection. Delta mutations E484R and P681R are gaining dominance, and this variant may be of high concern especially if these mutations are coupled with the N501R mutation (not emerged so far), which is a stronger binder compared to N501H of the Alfa variant. Arg, which is a highly hydrophilic, polar, and strongly positive amino acid, seems to be the critical key residue for the high affinity of SARS-CoV-2 with ACE2 and cleavage to S1 and S2 subunits prerequisite for the virus-cell entry and infection. Therefore, effective blockade of the spike affinity to ACE2 can be achieved by blocking the resulting arginine mutant residue via arginine blockers, and bisartans bearing two tetrazoles may be an ideal treatment [44]. A recent study has shown that bisartan BisD inhibits infection in bioassays, but not as effectively as PF-07321332 [14]. Figure 16 shows a close view of bisartan A bound to ACE2 close to the Zn-binding catalytic site of the receptor also R452 in S-protein RBD close to the nearest negatively charged residues (E22, D38, and E35) of ACE2 receptor. It is worth to report a recent excellent study on SARS-CoV-2 induced senescent cells as a fertile environment for viral mutagenesis, by Evangelou et al., Eur Respir J. 2022 [83].

## 5. Conclusions

Most mutations cited in this report resulted in destabilization of the native S-protein conformation, as well as a predicted reduction of RBD-ACE2 binding, as reflected by ddG and dG_bind_ free-energy perturbation calculations, respectively. Such a result should not be surprising, since it is a general tenet of biology that most mutations do not manifest pathogenic significance and, indeed, they often diminish viral fitness and persistence in host reservoirs. MD simulations presented herein of selected single-point mutant forms of the RBD (e.g., N501Y Alpha or L452R Epsilon), as well as strains harboring multiple RBD mutations (e.g., N501Y, K417N and E484K; Figure 6), qualitatively agree with the known (enhanced) pathogenicity of these variants in terms of their association with increased transmissibility and possibly disease severity. Indeed, some RBD mutations (e.g., N501Y), clearly create stronger links with ACE2 interfacial residues, which have been cited as major contributing factors to increased viral transmission [31]. The potential value of in silico methods is demonstrated by their ability to make informed predictions regarding the behavior of as yet undetected mutant forms of the RBD. For example, as shown in Figure 6, Figure 8 and Figure 9, the N501W mutation predicts an exceptional increase in the RBD-ACE2 binding strength, which could lead to more rapid spread of the virus. Yet, to our knowledge, this variant has not yet been detected in the human population. Additionally, results of SMD simulations presented herein suggests a “strong binding domain” (SBD) encompassing RBD residues 474–490 and *alpha*-helical ACE2 residues 19–28 and 78–83 (see Figure 8A and Figure 9). Recently, Chan et al. and Glasgow et al. [84,85] reported that lab-engineered ACE2 constructs harboring specific mutations exhibited significantly enhanced in vitro attachment to human SARS-CoV-2 S-proteins. It was demonstrated that such mutant constructs can be used as soluble “ACE2 traps” [85] to efficiently bind the virus and prevent infection [84]. It is perhaps noteworthy that two of the three ACE2 mutation loci (T27Y and L79T) that resulted in the greatest enhancement of RBD binding efficiency were located in the SBD region identified in our SMD simulations. The third ACE2 mutation (N330Y in the engineered ACE2 construct [84]) is located in the general vicinity of N501 of the RBD. Mutations at N501 (e.g., N501Y and N501W) have been identified in our study as being influential in enhancement of RBD-ACE2 interactions.

Our modeling results also suggest that the non-RBD P681R near to the furin arginine-rich cleavage site (positions 685–686) catalyzes the cleavage that creates the subunits S1 and S2 for the entry to ACE2, triggering infection. The aromaticity at position 501 plays an important role with increasing binding strength from Phenylalanine to Tyrosine and Tryptophan due to increased π-π interactions. Arg blockers, such as bisartans containing two tetrazoles to block RBD arginine mutants and Zn^2+^ ACE2 are potential drugs in the treatment of COVID-19. Bisartans, in particular, act at three targets essential for viral infection and replication (i.e., ACE2, furin, 3CLpro) and are promising candidates for clinical trials.

## Figures and Tables

**Figure 1 viruses-14-01029-f001:**
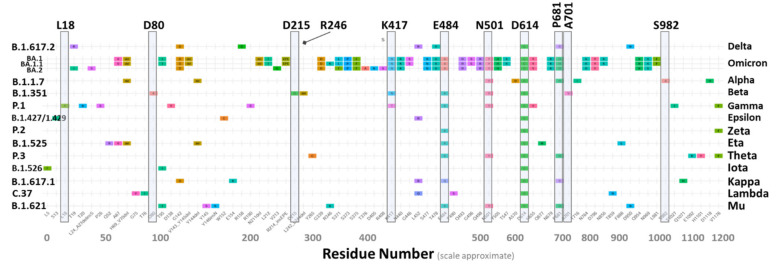
Selected SARS-CoV-2 dominating variants and mutations. Vertical shaded bars indicate variants examined in this investigation. Table modified from WHO Webpage (click on link for full-scale image): https://www.who.int/docs/default-source/coronaviruse/voc_voi_current_previous.pdf?sfvrsn=990a05c2_8.

**Figure 2 viruses-14-01029-f002:**
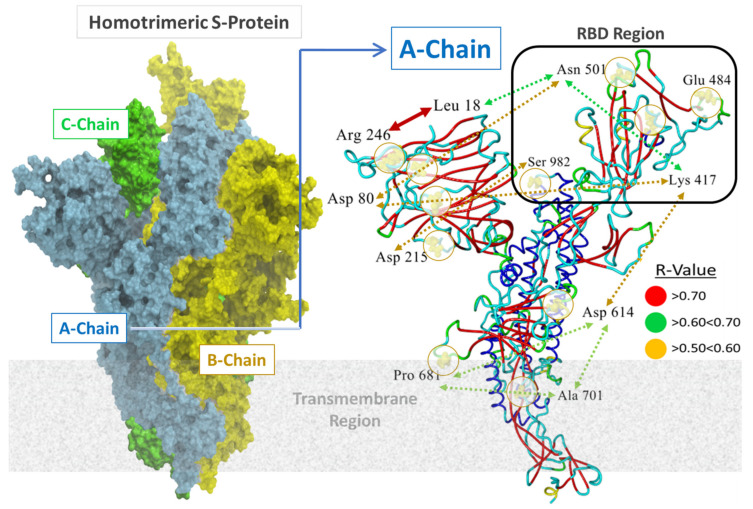
**Left Panel**: Surface rendition of the homology construct (Template = PDB 6XR8) of the SARS-CoV-2 S-glycoprotein employed in this study. This full-sequence model (SM#05; residues 14–1146) was constructed and refined in (ICM-Chemist_Pro software; Molsoft, LLC) based on the 3D electron microscopic structure found at PDB 6XR8 (SM ID = 6xr8.1) (see https://swissmodel.expasy.org/repository/species/2697049 for details, accessed on 28 April 2022). Sequence similarity was 0.62. The Global Model Quality Estimation value for this model was 0.73, indicative of a high-quality model. **Right Panel**: Locations of 11 residues of interest in this study (mutation loci) in the A-chain of Homotrimeric S-Protein are indicated. Double-headed arrows indicate residue pairs that were statistically correlated (R > 0.5) in free-energy protein stability calculations averaged over 20 amino acid substitutions at each locus. Higher correlations indicate mutations at each pair locus affect protein stability to a similar magnitude. For example, sites R246 and L18 exhibited the highest correlation (R = 0.87) suggesting the possibility that mutation-driven conformational changes at one site may influence conformation at the other site in a manner that also favors enhancement of S-protein stability.

**Figure 3 viruses-14-01029-f003:**
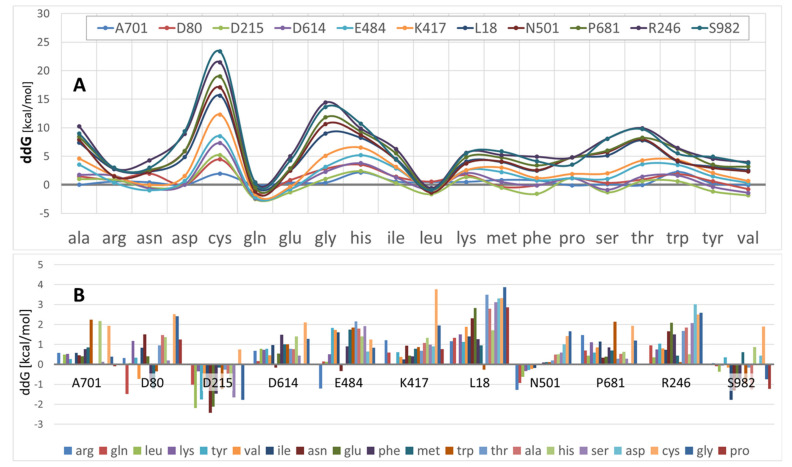
(**A**) Stacked line graphic comparing S-protein ddG across 11 single-point mutations (color-coded lines) for all 20 amino acid substitutions. Computed ddG values falling below “0” enhance proteo-stability of the S-protein. (**B**) Calculated ddG values as a function of mutation site (e.g., N501) and amino acid substitutions (color-coded). More positive ddG values reflect increasing protein instability. On average S-protein sites D80, D215, N501, and S982 resulted in enhanced proteo-stability, i.e., stability was enhanced at each of these sites for a larger proportion of the amino acid population. *Abbreviations:* Ala, alanine; Arg, arginine; Asn, asparagine; Asp, aspartic acid; Cys, cysteine; Gln, glutamine; Glu, glutamic acid; Gly, glycine; His, histidine; Ile, isoleucine; Leu, leucine; Lys, lysine; Met, methionine; Phe, phenylalanine; Pro, proline; Ser, serine; T, threonine; Trp, tryptophan; Tyr, tyrosine; Val, valine.

**Figure 4 viruses-14-01029-f004:**
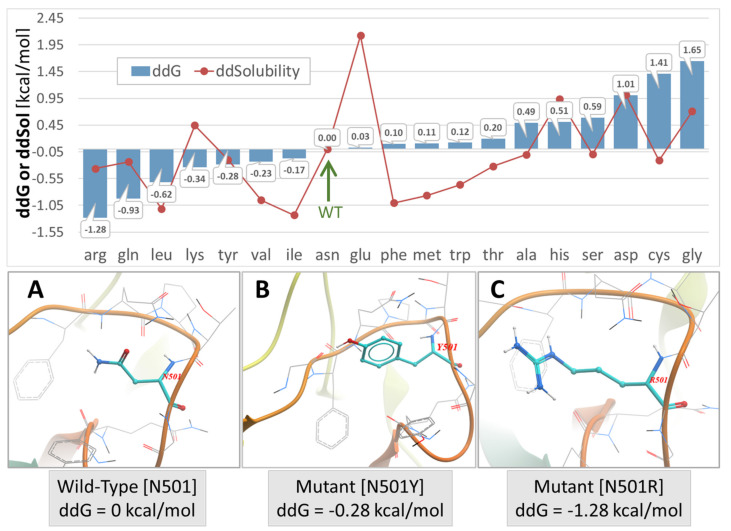
**Upper Panel**: S-protein (Swiss Model #05) stability expressed by change in ddG or free-energy of solubility (ddSol) plotted as a function of mutation type (amino acid substitution) at the N501x locus of PDB 7KGK. The calculated free-energy values for each mutation are expressed relative to the wild-type N501, which was assigned a value of zero. Lower values of ddG and ddSol correspond to increased protein stability. Thus, an Asn to Arg substitution at the N501 locus resulted the largest enhancement of protein stability (−1.28 kcal/mol). The green arrow indicates the wild-type N501 S-protein. **Lower Panel**: (**A**) Asn conformation at the wild-type N501 locus. (**B**) Mutant N501Y conformation. (**C**) Mutant N501R conformation. All conformers were modeled directly from the 6LZG x-ray structure using ICM-Chemist-Pro (Molsoft, LLC, San Diego, CA, USA). Atom Colors: cyan = carbon; blue = nitrogen; red/orange tubes = protein backbone.

**Figure 5 viruses-14-01029-f005:**
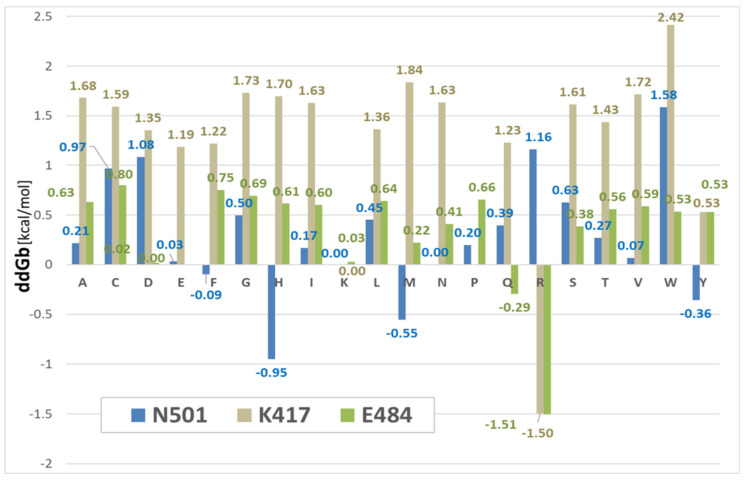
Calculated changes in the free-energy of binding [dG_bind_] between the SARS-CoV-2 RBD and the human ACE2 receptor for the PDB 6LZG complex. The calculations were performed for 20 amino acid substitutions (single-letter residue labels in the graphic) at the three specific mutation loci indicated, including: N501, K417, and E484. Interestingly, a N501H and N501M mutations resulted in enhancement of RBD-ACE2 binding that exceeded that of the known Alpha mutant form (N501Y).

**Figure 6 viruses-14-01029-f006:**
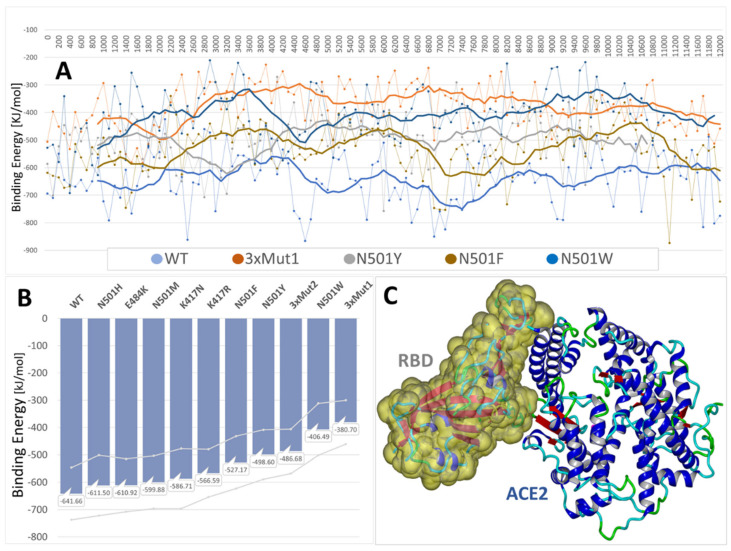
(**A**) Comparison of RBD-ACE2 binding energy MD trajectories for five RBD variants. For reasons of visual clarity, only five of the nine MD simulation trajectories are plotted. These included the wild-type strain (WT), the triple mutant “3xMut1” (see below), N501Y, N501F, and N501W. Heavy solid lines are 10-point rolling averages. Thinner lines are binding energy data calculated at 100-ps intervals. The wild-type PDB 6LZG complex and variants thereof were used to perform 12–18 ns of MD simulation in physiological saline at 311^O^K (NPT ensemble, AMBER-14; initial 12 ns are plotted). Simulations included the following RBD variants: (1) wild-type 6LZG, (2) N501H, (3) E484K, (4) N501M, (5) K417N, (6) K417R, (7) N501Y, (8) triple mutant “3xMut2” harboring N501Y, K417R, and E484K, and (9) triple mutant “3xMut1” harboring N501Y, K417N, and E484K. (**B**) Average binding energies (vertical bars) and standard deviations (gray lines) over the nine MD trajectories (*n* = 120–180). (**C**) RBD-ACE2 complex of 6LZG at t = 0 ns. The RBD fragment is depicted as the Van der Waals surface. Water and NaCl ions have been hidden for clarity.

**Figure 7 viruses-14-01029-f007:**
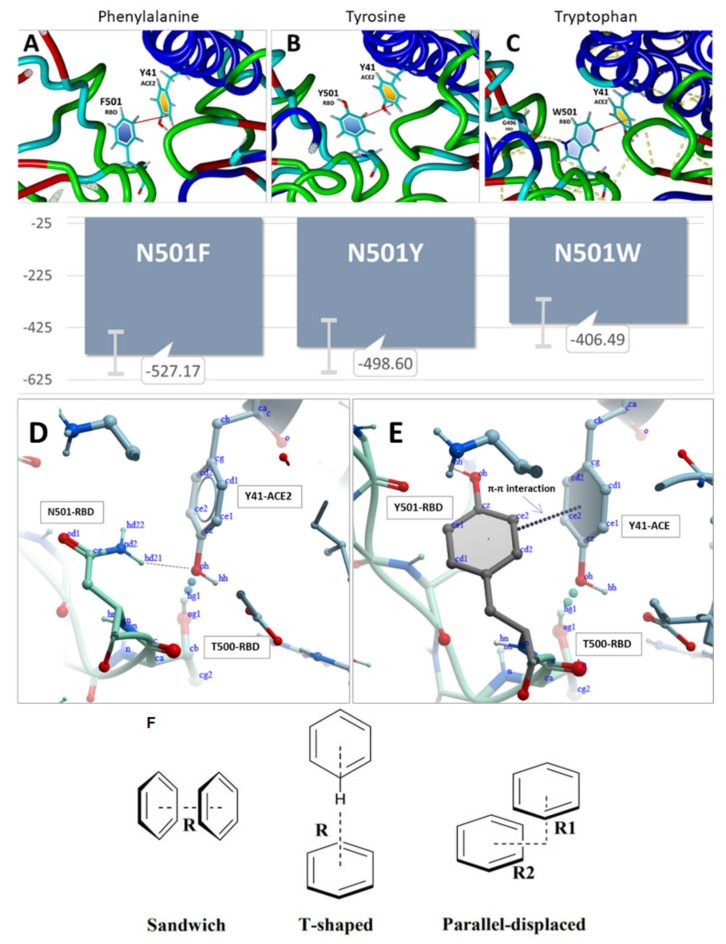
RBD-ACE2 binding energies (KJ/mol) derived from the MD simulations. N = approximately 120 in each case. Higher (less negative) energies correspond to stronger RBD-ACE2 binding. **Upper Panel**: Intermolecular π-π interactions (red lines) between Y41 of the ACE2 receptor and (**A**) phenylalanine (N501F), (**B**) tyrosine (N501Y), or (**C**) tryptophan (N501W) substitutions at the N501x position of the SARS-CoV-2 S-protein RBD (PDB entry 6LZG). Each RBD-ACE2 conformation shown was obtained following 9.0 ns of MD equilibration at 311^O^K in physiological saline. **Middle Panel**: (**D**) Hydrogen bonding between donor atom hd21 of N501 of the RBD and the phenyl oxygen acceptor (oh atom) of residue Y41 of the ACE2 receptor. (**E**) π-π interactions between Y41 of the ACE2 receptor and tyrosine (N501Y) of S-protein RBD. The distances between RBD Residue/Atom and ACE2 Residue/Atom may be viewed in the Appendix A. **Lower Panel**: (**F**) Sandwich, T-shaped, and parallel-displaced configurations of the benzene dimer.

**Figure 8 viruses-14-01029-f008:**
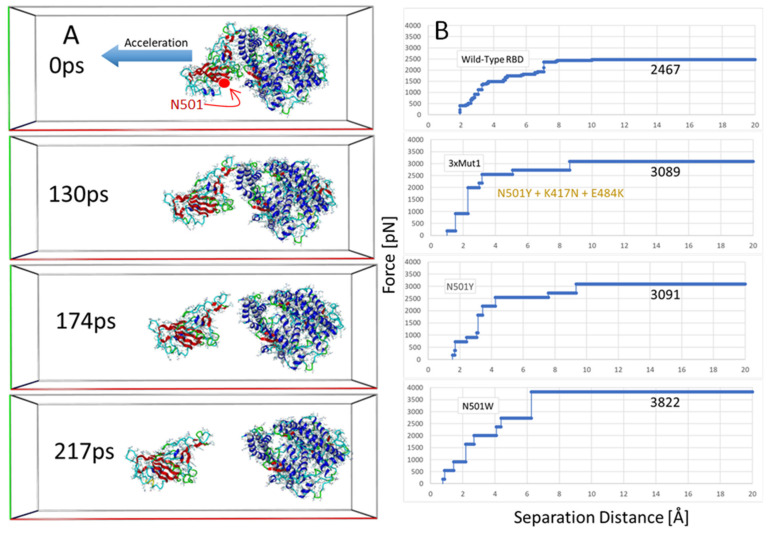
(**A**) Frame-captures from the SMD simulation of the wild-type RBD-ACE2 complex (6LZG) showing detachment of the RBD from the tethered ACE2 receptor. Water and NaCl ions have been hidden for clarity. Note that the lower part of the interfacial region near N501 residue locus of the RBD (red arrow) was the first area to undergo complete separation from the ACE2 receptor. The upper region of the interfacial zone furthermost from N501 was the last to undergo separation after about 160–170 ps of elapsed time. (**B**) Pulling force (pN) as a function of separation distance (Å) for the wild-type 6LZG complex (lower graph), RBD mutant N501W, RBD triple mutant 3xMut1, and RBD mutant N501W.

**Figure 9 viruses-14-01029-f009:**
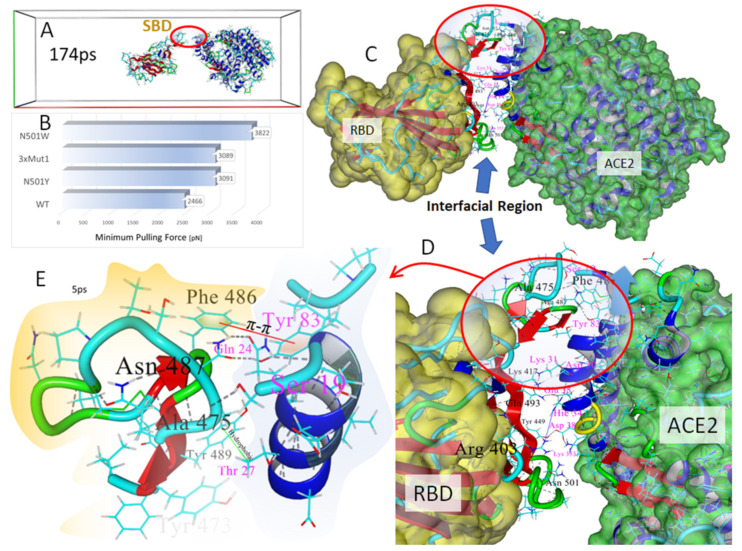
Interactions in the “strong binding domain” (SBD) of the wild-type RBD-ACE2 complex (6LZG). The SBD at the “top” of the interfacial region (circled in red) approximately comprised RBD residues 474–490 and ACE2 residues 19–22 and 78–83. The ACE2 SBD region is dominated by the two parallel *alpha*-helical regions that overlay the central zinc cavity of ACE2. (**A**) View of RBD-ACE2 complex at 174ps of SMD simulation time following near complete dissociation of the proteins. Note that the “strong binding domain” (SBD) was the last region to undergo dissociation. (**B**) Minimum pulling force (MPF) required for complete dissociation of the RBD-ACE2 complexes as a function of mutant type. (**C**) Wild-type RBD-ACE complex showing overview of interfacial zone after initial system equilibration and 104ps of SMD. RBD residue labels are shown as black; ACE2 labels as magenta. (**D**) Magnified view of interfacial region indicating significant intermolecular interactions, including: strong salt bridge from Arg402 of the RBD to Hie34 of ACE2 (blue line in (**D**)); weak salt bridge from Lys417 of the RBD to Asp30 (gray-blue line in (**D**)); π-π interaction between Phe486 of the RBD with Tyr83 of ACE2; hydrogen bond between the N-terminus Ser19 of ACE2 and Ala475 carbonyl oxygen of the RBD (dotted lines); hydrogen bond between RBD Tyr 449 -OH group and ACE2 Asp38; and hydrogen bond between the RBD Gln493 and ACE2 Glu35. (**E**) Detail view of some of the principle attractive interactions in the SBD. These include numerous stabilizing hydrogen bonds (dotted lines), as well as a strong π-π interaction between Phe486 of the RBD and Tyr83 of ACE2. Asn487 of the RBD was involved in two hydrogen bonds to the -OH group of Tyr83 of ACE2 and the terminal carbonyl group of Gln 24 of ACE2. A hydrophobic interaction exists between RBD Tyr489 and ACE2 Thr27. The hydrogen bond between Ala475 of the RBD and Ser19 of ACE2 helped stabilize SBD interactions and was often the last bond to undergo dissociation as the proteins were pulled apart.

**Figure 10 viruses-14-01029-f010:**
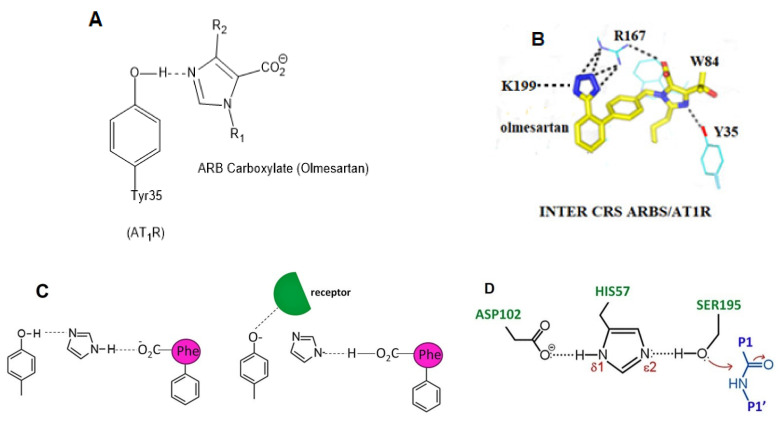
Intermolecular interactions between AT_1_R and the ARB, Olmesartan. (**A**) Explanatory graph depicting the hydrogen bond interaction between AT_1_R Tyr35 hydroxyl with Olmesartan imidazole nitrogen. (**B**) Interactions of AT_1_R residues K199, R167, W84, Y35 with olmesartan from crystal protein bank [45,46]. **Note.** ARBs may be inhibitors of SARS-CoV-2 and ACE2 binding through strong interactions between negatively charged tetrazolate and carboxylate with polybasic arginines cavity loop 681–686 S protein RBD of SARS-CoV-2. (**C**) Tyrosine (Tyr), histidine (His), and phenylalanine (Phe) aromatic side chains and c-terminal carboxylate are intra connected to create a cyclic compact structure that triggers activity through Tyr hydroxylate. (**D**) His and asparagine (Asp) carboxylate in serine proteases are intra connected to create the CRS in serine proteases. (**E**) Explanatory graph depicting inter molecular interactions between EXP3174 negative groups histidine carboxylate and biphenyl tetrazolate with positively charged AT_1_R residue arginine 167. Tyr35 hydroxylate of AT_1_R interacts with biphenyl imidazole nitrogen. (**F**) Explanatory graph depicting intra molecular interactions between AngII with AT_1_R167 and Tyr35. **Note**: The interactions between AT_1_R Arg167 with His carboxylate and biphenyl tetrazolate can occur in silico between SARS-CoV-2 basic loop Args with carboxylate and tetrazolate of ARBs.

**Figure 11 viruses-14-01029-f011:**
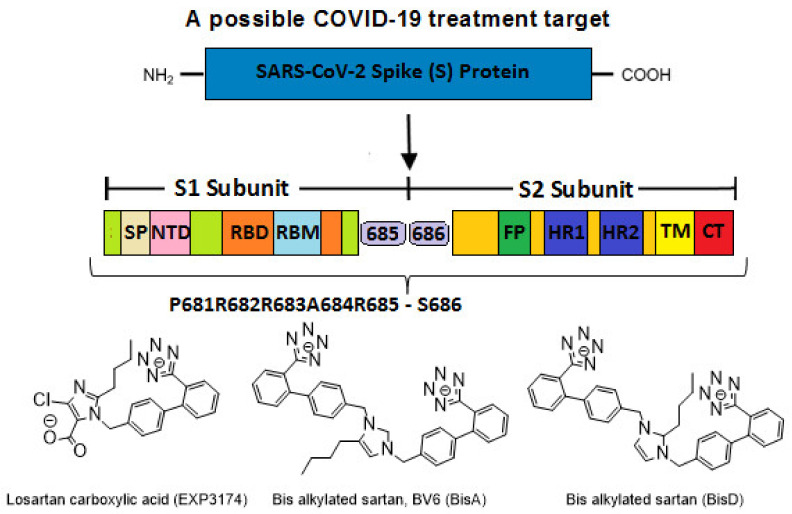
Structures of losartan carboxylic acid (EXP3174), BisA and BisD. The multibasic cleavage site of SARS-CoV-2 (P681R682R683A684R685S686) is a treatment target for negatively charged groups or molecules such as ARBs that possibly target the basic cleavage site, preventing furin cleavage which triggers infection. SARS-CoV-2 Spike (S) Protein consists of two subunits, the surface subunit (S1) and a membrane-anchored subunit (S2). The S1 subunit contains an N-terminal signal peptide (SP), an N-terminal domain (NTD) a receptor-binding domain (RBD) and inside that the receptor-binding motif (RBM). The S2 subunit comprises domains required for membrane fusion, the fusion peptide (FP), two heptad repeats (HR1, HR2), a transmembrane domain (TM) and a cytoplasmic tail (CT). The S1/S2 cleavage site is located at the border between the S1 and S2 subunits (R685-S686).

**Figure 12 viruses-14-01029-f012:**
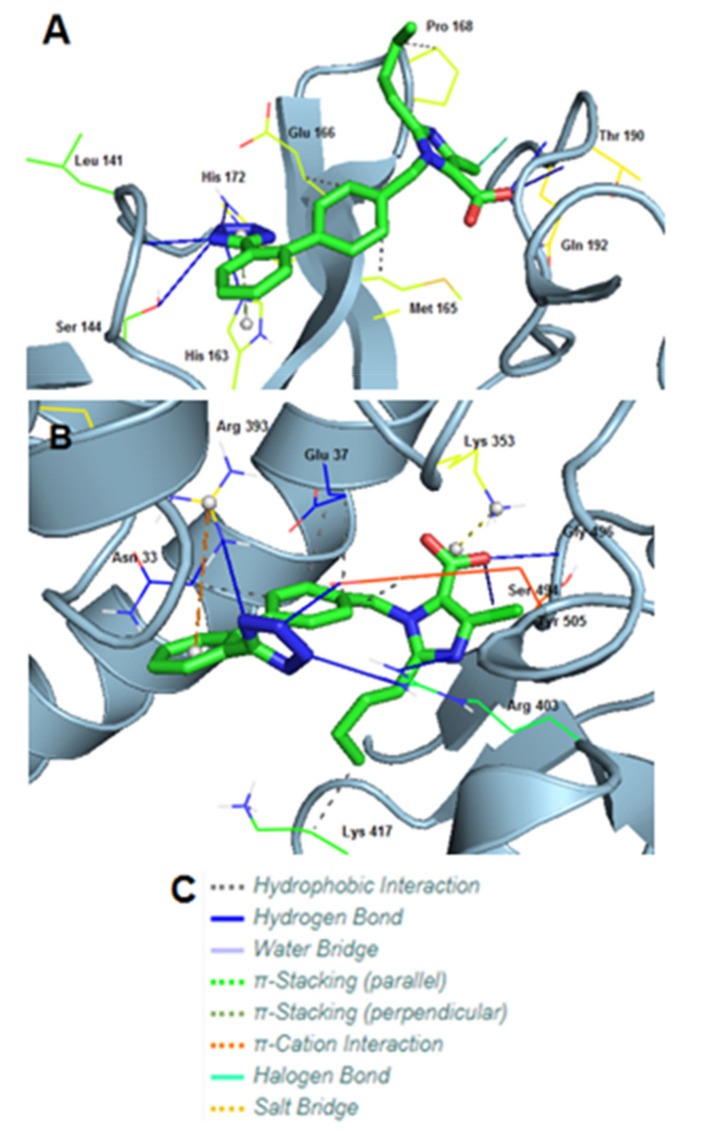
(**A**) EXP3174 incorporated in a cavity of Mpro of SARS-CoV-2 (PDB entry code: 6LU7), using crystallographic grid. (**B**) EXP3174 incorporated in a cavity of S-protein RBD (PDB entry code: 6LZG), of SARS-CoV-2 using crystallographic grid. (**C**) Depiction of the various protein-ligand interactions, according to PLIP.

**Figure 13 viruses-14-01029-f013:**
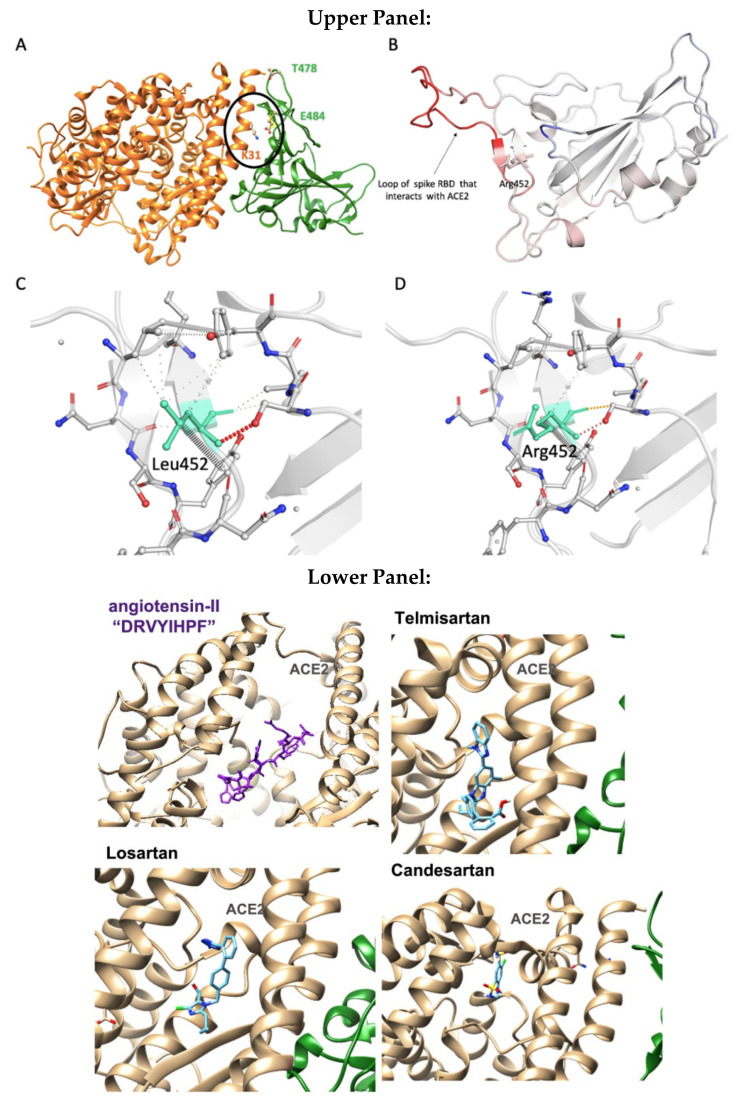
**Upper Panel:** (**A**) Contacts between Lys31(ACE2) and Glu484 (S-protein RBD) in the human-virus interface. (**B**) Amino acids of S-protein RBD are colored according to the vibrational entropy change upon L452R mutation: blue represents a rigidification of the structure and red: a gain in flexibility. (**C**,**D**). Wild-type (L452) and mutant residue (R452) are colored in light-green and represented as sticks alongside with the surrounding residues which are involved on any type of interactions. **Lower Panel:** The binding site of AngII (purple), telmisartan, losartan, candesartan in the ACE2 zinc-domain groove (light gold) complexed with S-protein RBD (green).

**Figure 14 viruses-14-01029-f014:**
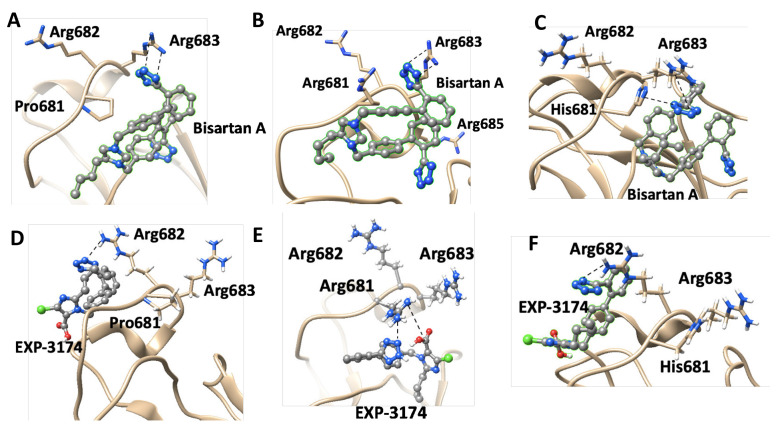
**Upper panel:** Closed view of docking pose of bisartan A in furin cleavage site of wild-type ACE2 (**A**), P681R ACE2 (**B**) and P681H ACE2 (**C**). Lower panel: closed view of docking pose of EXP-3174 in furin cleavage site of wild-type ACE2 (**D**), P681R ACE2 (**E**) and P681H ACE2 (**F**).

**Figure 15 viruses-14-01029-f015:**
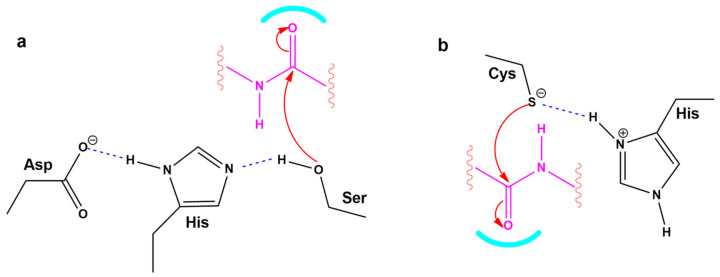
CRS Protease mechanisms. SARS-CoV-2 S protein can be cleaved by furin at positions 685–686 and 3CLpro at glutamine positions through CRS mechanisms. The catalytic center of furin is the triad Asp—His—Ser, and for 3CL protease is the dyad Cys145—His41. Two of the four main mechanistic classes of proteases are (**a**) serine proteases as in the furin cleavage, and (**b**) cysteine proteases as in the 3CLpro hydrolysis [53].

**Figure 16 viruses-14-01029-f016:**
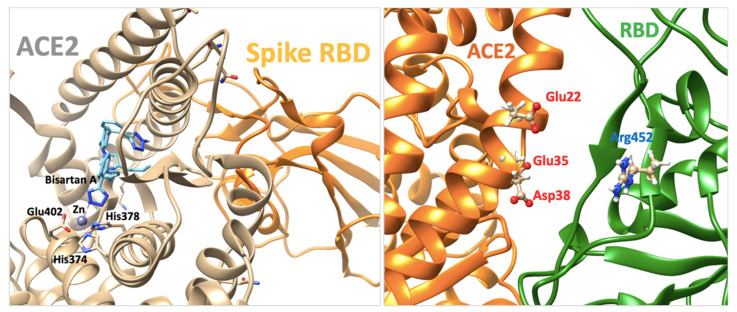
**Left**: Close view of bisartan A bound to ACE2, close to the Zn-binding catalytic site of the receptor. **Right**: R452 in S-protein RBD close to the nearest negatively charged residues (E22, D38, and E35) of ACE2 receptor.

**Table 1 viruses-14-01029-t001:** Computed binding energies (in kcal/mol) of AngII and sartans on S-protein RBD-ACE2 complex. The higher the negative number the higher the binding energy. Binding energies and affinity constants should be viewed in relative terms—they highlight differences between ligands and do not account for entropy differences.

Ligand	Best Scoring Pose Bound to ACE2-Spike RBD Complex (kcal/mol)
Candesartan	−7.1
Losartan	−8.1
Telmisartan	−9.5
BisA	−9.5
AngII	−12.4

**Table 2 viruses-14-01029-t002:** According to the Kyte-Doolittle scale (J. Mol. Biol. 1982), there are three IMGT ‘hydropathy’ classes [29]. Amino acids in each class are ordered from the most hydrophobic on the left-hand side (i.e., I) to the most hydrophilic on the right-hand side (i.e., R) [29]. *Abbreviations:* A, alanine; R, arginine; N, asparagine; D, aspartic acid; C, cysteine; Q, glutamine; E, glutamic acid; G, glycine; H, histidine; I, isoleucine; L, leucine; K, lysine; M, methionine; F, phenylalanine; P, proline; S, serine; T, threonine; W, tryptophan; Y, tyrosine; V, valine.

I	V	L	F	C	M	A	W	G	T	S	Y	P	H	N	D	Q	E	K	R
4.5	4.2	3.8	2.8	2.5	1.9	1.8	−0.9	−0.4	−0.7	−0.8	−1.3	−1.6	−3.2	−3.5	−3.5	−3.5	−3.5	−3.5	−4.5
Hydrophobic								Neutral						Hydrophilic					

**Table 3 viruses-14-01029-t003:** Interactions of amino acid residues with EXP3174 at receptor sites of MPro and RBD of S protein of SARS-CoV-2 using PLIP. Binding energies and affinity constants should be viewed in relative terms—they highlight differences between ligands and do not account for entropy differences.

Type of Interactions	Amino Acids of Mpro Involved and Distance of Interactions (A°)	
Hydrogen Bonds	Leu141	1.91
	Ser144	3.40
	His163	2.00
	His172	3.61
	Thr190	2.09
	Thr190	1.86
	Gln192	1.98
Hydrophobic Interactions	Met165	3.25
	Glu166	3.97
	Pro168	3.82
π-Stacking (Stacking Type T)	His163	4.11
Halogen Bonds	Gln192	3.61

**Table 4 viruses-14-01029-t004:** Interactions of amino acid residues with EXP3174 at receptor site of RBD of S protein of SARS-CoV-2 using PLIP.

Type of Interactions	Amino Acids of S-RBD Involved and Distance of Interactions (A°)	
Hydrogen Bonds	Arg393	2.47
	Arg403	2.42
	Arg403	2.12
	Ser494	3.38
	Gly496	1.83
	Tyr505	1.94
Hydrophobic Interactions	Asn33	3.94
	Glu37	3.75
	Glu37	3.45
	Lys417	3.63
	Tyr505	3.50
π-Cation Interactions	Arg393	5.33
Salt Bridges	Lys353	2.73

## Data Availability

Not applicable.

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
