# Peer review of "Understanding the Driving Forces That Trigger Mutations in SARS-CoV-2: Mutational Energetics and the Role of Arginine Blockers in COVID-19 Therapy"

_viruses, 2022, doi:10.3390/v14051029_

Round 1

Reviewer 1 Report

This in silico study found the driving force triggered mutations in SARS-CoV-2 and suggested replacement of non-polar hydrophobic interactions with polar hydrophilic interactions such as Arginine might enhance binding of RBD with ACE2. My remarks and questions are listed as follows:

Major reviews:

1.Several critical amino acid of mutations on the interface between RBD and ACE2 were identified via silico study, however, these had been widely verified through experimental researches already.

2.New mutation prophecy such as N501R were promoted in this paper, however, frankly no VOCs showed clear trends of mutation other than N501Y, so as to K417 and E484 so far, hence, more data and analysis should be performed in molecular epidemiology level to prove your identification or to retrieve previous evidence of this possibility.

3.Try to explain the origin and identification of your novel drug, we still confused about the relationship among bisartan, bisartan A, EXP3174 and other chemical compounds.

4.The text did not align with the figure. It was quite confusing which figure showed is stated for which purpose, and more problems for the layout of this paper should be treated carefully indeed as depicted in the following minor reviews.

Minor reviews:

  1. Many figures like Table2,Figure3D.E, Figure4B,their resolution were too low for publishing, hence, suggesting for substitution, so as to problems like titles, figures and tables should be modified under same standard (font, size, figure shapes all included).
  2. Strongly suggest the author to move some parts of the table to the supplementary materials.
  3. More background and emphasis for the purpose and scientific question that your study could resolve.
  4. In line 144, it should be “RBD-ACE2 complex”.
  5. It is noticed that in line 320, F, Y, W mutation were all aromatic amino acids, is there any chances this trait contributed to the strongly binding affinity, and should there should be evidences to prove these mutations did not affect much to the original conformation of the loop, since this mutation changed the polarity of the amino acids.
  6. Visualized structure figure 7 was strongly suggested to be rayed and present with higher resolution, and provide more detailed interactions in the figures.
  7. In line 345, it is better to clarify the information of the origin and function of EXP3174.
  8. Omicron was strongly suggested to be discussed around line 436.
  9. Some of the discussion could be paraphrased in the introduction and result part for description(e.g. line 513-538 for introduction,and line573-622 for results).
  10. In line 502, in omicron some of the research in omicron shown no enhancement of cleavage activity, which should be explained.

Author Response

We are grateful to the reviewer for constructive comments and suggestions. We believe we have complied with their suggestions. We believe we have complied with the suggestions. We also reconstructed the paper to include sections 3.5, 3.6, 3.7, 3.8, and three new figures 1, 7, 8. Title is also changed to point out the mutational energetics.

Reviewer 1
This in silico study found the driving force triggered mutations in
SARS-CoV-2 and suggested replacement of non-polar hydrophobic
interactions with polar hydrophilic interactions such as Arginine might
enhance binding of RBD with ACE2. My remarks and questions are listed as
follows:

Major reviews:

1.Several critical amino acid of mutations on the interface between RBD
and ACE2 were identified via silico study, however, these had been
widely verified through experimental researches already.

Reply Indeed several amino acid mutations on the interface between RBD and ACE2 as well non RBD mutations have been identified to be critical for transmissibility and severity. Our in silico MD study attempts to shed light to the atomic properties   that direct mutations. We believe this study has further enlightened the driving forces that trigger mutations.     2.New mutation prophecy such as N501R were promoted in this paper,
however, frankly no VOCs showed clear trends of mutation other than
N501Y, so as to K417 and E484 so far, hence, more data and analysis
should be performed in molecular epidemiology level to prove your
identification or to retrieve previous evidence of this possibility.   Reply. In silico studies scrutinize the molecular and atomic properties of the wild  and mutated variants and can explain the transformation and the ability of mutants to strongly bind to the ACE2. These studies show that a possible N501R mutation could tight stronger with ACE 2 in line with the hydropathy polarity of the residues. No such mutation was luckily observed so far which could be maybe fatal especially if combined with  L452R and E484Q.

3.Try to explain the origin and identification of your novel drug, we
still confused about the relationship among bisartan, bisartan A,
EXP3174 and other chemical compounds.   Reply As suggested the structures of EXP3174( Losartan Carboxylic acid) and Bisartan  BisA and their structural differences are clearly shown in corrected Fig 5. We were interested also in this study to investigate Sartans as possible RBD /ACE2 blockers due to their ability with their negative tetrazole to neutralise  positive guanidino group of arginine as in critical mutations  like RBD L452R and non RBD P681R. We have found that bisartans bearing two tetrazoles as BisA are stronger binders compared to known Sartans as Losartan and Exp3174( Losartan Carboxylic) bearing only one tetrazole or combination of tetrazole / carboxyl. Bisartans block in silico efficiently arginine mutants and also the  rich arginine catalytic cleavage site 681-686. Their binding properties are clearly described in Discussion Section  entitled “ Bisartans and the role of tetrazole”. (new ref 14 is also included)

4.The text did not align with the figure. It was quite confusing which
figure showed is stated for which purpose, and more problems for the
layout of this paper should be treated carefully indeed as depicted in
the following minor reviews.   Reply  We checked that  figures  are lined up with text correctly. Figure legends describe and explain in detail related statements in the text.   All parts referring to Omicron are deleted as confusing and not contributing to the message.   Omicron mutations, not clinically severe, could not be analysed  as the number is high( 30 mutations). It is suggested that a conformational change should affect the RBD /ACE2 complex, not possible to analyse on the Omicron variant.
Minor reviews:

Many figures like Table2,Figure3D.E, Figure4B,their resolution were too
low for publishing, hence, suggesting for substitution, so as to
problems like titles, figures and tables should be modified under same
standard (font, size, figure shapes all included).   Reply by  Resolution was increased in figures.  
Strongly suggest the author to move some parts of the table to the
supplementary materials.   Reply. A new table 1 is included which shows all variants with dominating mutations.  
More background and emphasis for the purpose and scientific question
that your study could resolve.   Reply Background and emphasis to the purpose of understanding the driving forces of mutations is extensively described in Introduction and Discussion. Furthermore, this study suggests treatment targets and arginine blockers. 
In line 144, it should be “RBD-ACE2 complex”.   Reply It is RBD-ACE2 complex.  
It is noticed that in line 320, F, Y, W mutation were all aromatic amino
acids, is there any chances this trait contributed to the strongly
binding affinity, and should there should be evidences to prove these
mutations did not affect much to the original conformation of the loop,
since this mutation changed the polarity of the amino acids.   Reply We don’t have solid evidence for the conformation before and after the mutation. Some conformational changes certainly  occur. Obviously the pi-pi factor which increases in the order F,Y,W is the critical one contributing to stronger RBD /ACE2 complex. This is suggested and shown in the in silico studies (Figure 3).
Visualized structure figure 7 was strongly suggested to be rayed and
present with higher resolution, and provide more detailed interactions
in the figures.   Reply by Figure is presented with higher resolution.
In line 345, it is better to clarify the information of the origin and
function of EXP3174.   Reply Exp3174 is the oxidative product of Losartan where methylene hydroxyl group is oxidised enzymatically to Carboxylic acid providing Losartan carboxylic acid. This is clarified within the Discussion section and Fig 5.     Omicron was strongly suggested to be discussed around line 436.   Reply Omicron report and discussion is deleted  through the text as confusing. No solid data are available to explain moderate transmissibility of Omicron. We delete last paragraph in Introduction relevant to Omicron.
Some of the discussion could be paraphrased in the introduction and
result part for description(e.g. line 513-538 for introduction,and
line573-622 for results).   Reply As suggested we paraphrased and shortened description within lines 513-538  and 573-622.  
In line 502, in omicron some of the research in omicron shown no
enhancement of cleavage activity, which should be explained.   Reply Omicron statement is deleted as it is confusing.

Reviewer 2 Report

General comments

The authors performed in-silico analysis of different complexes formed between SARS-CoV-2 S proteins, ACE2 receptor and chemicals/drugs. It was a good try and provided different explanations of current circulating SARS-CoV-2 variants. It focused on the atomic structures level. The methods used and results presented were appropriate, however, the manuscript was difficult to follow in terms of the manuscript organization, specific terms used, etc. The journal covers wide ranges of readers. It means that different backgrounds of the readers will get access to different topics. I am sure most of the readers of this journal are virologists. Although they are interested to this topic, they will not be able to follow the messages from the authors without knowing the backgrounds of each topic. Only a small proportion of the audiences are capable of going through this manuscript. The authors have to briefly introduce each concept and then go into details one by one. To proceed the revision, the authors are suggested to work with colleagues of medical field.

Specific comments

- the title of the manuscript is not appropriate: the authors analysed some spike mutations based on the known SARS-CoV-2 strains, although the authors used different models to realize the properties of not yet identified or rate mutations (e.g. 501F, 501W, 501R), the significance of these mutations remain unknown. The significance of Arginine blockers may work only on some SARS-CoV-2 variants, namely, Delta. This observation was based on the two significant mutations in spike, L452R and P681R. In addition, the trend of mutations from non-polar to polar may not be the only factor, indeed, the authors admitted that other factors (e.g. steric hindrance) contribute. However, for the unusual high number of mutations found in Omicron, the hypothesis may not work in this case. The title should be more neutral to reflect the experiments performed in the study.

My suggestion: In-silico atomic structure analysis of different complexes formed between SARS-CoV-2 spike proteins, ACE2 receptor and potential therapeutic drugs

- Below are the examples so that the manuscript can be improved. I cannot list them one by, however, the authors have to go through the manuscript and assess carefully.

(1) lines 58-66: regarding VOCs and VOIs, the reference should be included WHO.

(2) lines 73-76: it first introduced the actions of the chemical, the authors are suggested to elaborate ‘sartans’ and its interaction with amino acid using figure. Alternatively, the authors can either delete them or move to the paragraph of line 160.

(3) Table 1: you have to write down the criteria of showing amino acids here (e.g. D614G are present in all current circulating SARS-CoV-2, however, it was only shown in some variants)

(4) line 97: define AT1 and ARB

(5) line 100: I do not understand the self-contradict sentence ‘…… this was the most stabilizing mutation and the less stabilizing mutation……’

(6) line 160: the authors have to briefly introduce the chemicals/drugs first, EXP-3174, Olmesartan, Bisartans and then discuss them one by one.

(7) Figure 4E, F: need to increase the resolution, I cannot see the chemical bonds involved.

(8) line 215: at the end of introduction section, the authors have to summarize the aims of the study

(9) line 219: delete the word ‘point’

(10) Narrow down the objectives for the spike mutations analysed: 417, 484, 501, 452, 478, 681

(11) lines 307-332: need to break down into different sub-sections

- Discussion section

It is ok to separate each issue by using subheadings. However, they were too long, only need to discuss the results presented. You do not need to summarize other researches’ comprehensively. Only few sentences will work. This manuscript was not a review in nature.

Author Response

We are grateful to the reviewer for constructive comments and suggestions. We believe we have complied with their suggestions. We believe we have complied with the suggestions. We also reconstructed the paper to include sections 3.5, 3.6, 3.7, 3.8, and three new figures 1, 7, 8. Title is also changed to point out the mutational energetics.  

Reviewer 2
The authors performed in-silico analysis of different complexes formed
between SARS-CoV-2 S proteins, ACE2 receptor and chemicals/drugs. It was
a good try and provided different explanations of current circulating
SARS-CoV-2 variants. It focused on the atomic structures level. The
methods used and results presented were appropriate, however, the
manuscript was difficult to follow in terms of the manuscript
organization, specific terms used, etc. The journal covers wide ranges
of readers. It means that different backgrounds of the readers will get
access to different topics. I am sure most of the readers of this
journal are virologists. Although they are interested to this topic,
they will not be able to follow the messages from the authors without
knowing the backgrounds of each topic. Only a small proportion of the
audiences are capable of going through this manuscript. The authors have
to briefly introduce each concept and then go into details one by one.
To proceed the revision, the authors are suggested to work with
colleagues of medical field.

Specific comments

- the title of the manuscript is not appropriate: the authors analysed
some spike mutations based on the known SARS-CoV-2 strains, although the
authors used different models to realize the properties of not yet
identified or rate mutations (e.g. 501F, 501W, 501R), the significance
of these mutations remain unknown. The significance of Arginine blockers
may work only on some SARS-CoV-2 variants, namely, Delta. This
observation was based on the two significant mutations in spike, L452R
and P681R. In addition, the trend of mutations from non-polar to polar
may not be the only factor, indeed, the authors admitted that other
factors (e.g. steric hindrance) contribute. However, for the unusual
high number of mutations found in Omicron, the hypothesis may not work
in this case. The title should be more neutral to reflect the
experiments performed in the study.

My suggestion: In-silico atomic structure analysis of different
complexes formed between SARS-CoV-2 spike proteins, ACE2 receptor and
potential therapeutic drugs Reply As suggested the title is changed to be shorter: Understanding the driving forces that trigger mutations in SARS-CoV-2: The Role of Arginine Blockers in Covid-19 Therapy Also we deleted any description about Omicron as it is confusing and not possible to analyze due to the many mutations of this variant.

- Below are the examples so that the manuscript can be improved. I
cannot list them one by, however, the authors have to go through the
manuscript and assess carefully.

(1) lines 58-66: regarding VOCs and VOIs, the reference should be
included WHO.

Reply Reference 7 defines WHO
(2) lines 73-76: it first introduced the actions of the chemical, the
authors are suggested to elaborate ‘sartans’ and its interaction with
amino acid using figure. Alternatively, the authors can either delete
them or move to the paragraph of line 160.   Reply As suggested we moved the paragraph of lines 73-76 to the paragraph of line 160.

(3) Table 1: you have to write down the criteria of showing amino acids
here (e.g. D614G are present in all current circulating SARS-CoV-2,
however, it was only shown in some variants)

Reply As suggested by reviewer 1 we have left in table 1 the main variants Alpha and Delta. The rest were removed to supplementary material.
(4) line 97: define AT1 and ARB   Reply AT1 and ARB are defined in keywords

(5) line 100: I do not understand the self-contradict sentence ‘…… this
was the most stabilizing mutation and the less stabilizing mutation……’

Reply We added "N501Y mutation" in line 100 to clarify the statement and remove confusion.
(6) line 160: the authors have to briefly introduce the chemicals/drugs
first, EXP-3174, Olmesartan, Bisartans and then discuss them one by one.   Reply We have introduced Sartans with their pharmacophore groups at the end of Materials and Methods section as follows: Losartan is an angiotensin receptor blocker bearing tetrazole and hydroxyl methylene groups. EXP3174 is Losartan carboxylic acid bearing a tetrazolate and carboxylate. Olmesartan is Losartan analogue bearing a tetrazolate and a carboxylate. Bisartans are imidazole bis dialkylated biphenyl tetrazoles with the imidazole group bearing alkyl group at position 2 (Losartan-like) or 4 (Bisartan A).

(7) Figure 4E, F: need to increase the resolution, I cannot see the
chemical bonds involved.   Reply Resolution was increased.

(8) line 215: at the end of introduction section, the authors have to
summarize the aims of the study

Reply As suggested at the end of introduction section we summarize aim of the study. "The aim of this in silico study is to shed light to the driving forces that trigger mutations in SARS-CoV-2 strains and that increase transmissibility through enhanced interactions between RBD and ACE2 receptor and hydrolysis of the furin and 3CLpro cleavage sites."
(9) line 219: delete the word ‘point’   Reply The word point is correct.

(10) Narrow down the objectives for the spike mutations analysed: 417,
484, 501, 452, 478, 681   Reply The objective of the in silico study is furthermore to find blockers that prevent mutations 417, 484, 501, 452, 478, 681 which trigger and enhance infection. This objective is stated in the Abstract and Conclusions.
(11) lines 307-332: need to break down into different sub-sections
Reply This part was breakdown into subsections.

- Discussion section

It is ok to separate each issue by using subheadings. However, they were
too long, only need to discuss the results presented. You do not need to
summarize other researches’ comprehensively. Only few sentences will
work. This manuscript was not a review in nature.   Reply Subheadings were shortened as suggested.

Round 2

Reviewer 1 Report

The authors have made revisions based on the suggestions and the answers can be explained clearly, which meet the requirements of article publication.

Reviewer 2 Report

The authors addressed all of my queries.